# A RASSF1A-HIF1α loop drives Warburg effect in cancer and pulmonary hypertension

Swati Dabral[1], Christian Muecke[1], Chanil Valasarajan[1], Mario Schmoranzer[1], Astrid Wietelmann[2], Gregg L. Semenza[3], Michael Meister[4,5], Thomas Muley[4,5], Tamina Seeger-Nukpezah[6], Christos Samakovlis[1,7,8], Norbert Weissmann[8], Friedrich Grimminger[8], Werner Seeger[1,8], Rajkumar Savai[1,8] & Soni S. Pullamsetti[1,8]

Hypoxia signaling plays a major role in non-malignant and malignant hyperproliferative diseases. Pulmonary hypertension (PH), a hypoxia-driven vascular disease, is characterized by a glycolytic switch similar to the Warburg effect in cancer. Ras association domain family 1A (RASSF1A) is a scaffold protein that acts as a tumour suppressor. Here we show that hypoxia promotes stabilization of RASSF1A through NOX-1- and protein kinase C- dependent phosphorylation. In parallel, hypoxia inducible factor-1 α (HIF-1α) activates RASSF1A transcription via HIF-binding sites in the RASSF1A promoter region. Vice versa, RASSF1A binds to HIF-1α, blocks its prolyl-hydroxylation and proteasomal degradation, and thus enhances the activation of the glycolytic switch. We find that this mechanism operates in experimental hypoxia-induced PH, which is blocked in RASSF1A knockout mice, in human primary PH vascular cells, and in a subset of human lung cancer cells. We conclude that RASSF1A-HIF-1α forms a feedforward loop driving hypoxia signaling in PH and cancer.

[1] Department of Lung Development and Remodeling, Member of the German Center for Lung Research (DZL), Max-Planck-Institute for Heart and Lung Research, Bad Nauheim 61231, Germany. [2] MRI and μCT Service Group, Max-Planck-Institute for Heart and Lung Research, Bad Nauheim 61231, Germany. [3] Departments of Pediatrics, Medicine, Oncology, Radiation Oncology, Biological Chemistry, and Genetic Medicine, Johns Hopkins University School of Medicine, Baltimore MD21205 MD, USA. [4] Translational Research Unit, Thoraxklinik at Heidelberg University Hospital, Heidelberg 69126, Germany. [5] Translational Lung Research Center Heidelberg (TLRC-H), Member of the German Center for Lung Research (DZL), Heidelberg 69120, Germany. [6] Department I of Internal Medicine and Center for Integrated Oncology, University of Cologne, Cologne 50937, Germany. [7] Department of Molecular Biosciences, The Wenner-Gren Institute, Stockholm University, S-10691 Stockholm, Sweden. [8] Department of Internal Medicine, Universities of Giessen and Marburg Lung Center (UGMLC), ECCPS, Member of the DZL, Justus-Liebig University, Giessen 35392, Germany. Correspondence and requests for materials should be addressed to S.S.P. (email: soni.pullamsetti@mpi-bn.mpg.de)

Ras association domain family 1A (RASSF1A) is a scaffold protein modulating multiple apoptotic and cell cycle checkpoint pathways by facilitating assembly of multiple effector protein complexes[1]. Epigenetic silencing (promoter methylation) and genetic changes (somatic mutations) are observed in various cancers and in particular human cancer cells[2–4], establishing RASSF1A as a bonafide tumor-suppressor[5]. However, a few studies reported an increase in RASSF1A expression in primary human cancer tissues of various origin[6,7]. Further, the molecular mechanisms underlying RASSF1A regulation and function in primary human non-malignant cells are largely elusive. In addition, how this protein is regulated via environmental cues such as hypoxia remains unknown.

Hypoxia, defined as a reduction in the amount of oxygen available to a cell, tissue, or organism, is a fundamental and life-threatening biological phenomenon, and organisms from protozoans to complex mammals have evolved intricate mechanisms to sense changes in oxygen levels that in turn allow physiologic adjustments[8]. Hypoxic signaling pathways are implicated in a plethora of physiological processes such as blood cell differentiation and organ morphogenesis[9,10] Moreover, they are centrally involved in both malignant and non-malignant hyperproliferative disease processes. This is since long known as Warburg effect driving a fundamental metabolic (glycolytic) switch in the cancer cells[11] and is also operative in the prototype non-malignant hypoxia-driven disease, pulmonary hypertension characterized by vascular remodeling due to hyperproliferation of pulmonary vascular smooth muscle cells and adventitial fibroblasts[12]. A central axis of hypoxic signaling is the activation of the transcription factor hypoxia inducible factor-1 (HIF-1)[8]. HIF-1 consists of an oxygen regulated HIF-1α subunit and a constitutively expressed HIF-1β subunit. In oxygenated cells, HIF-1α is hydroxylated on two proline residues (Pro402 and Pro564 in human HIF-1α) by prolyl hydroxylases (PHD), leading to its rapid proteasomal degradation. Under hypoxic conditions, prolyl hydroxylation of HIF-1α is inhibited, leading to its stabilization and nuclear translocation. In the nucleus, HIF-1α dimerizes with HIF-1β and binds to the consensus sequence 5′-RCGTG-3′ embedded within hypoxia response elements (HREs) of numerous target genes[13]. Downstream gene regulation promotes key adaptive mechanisms like glycolysis and angiogenesis, but also drives pro-survival signaling, cell proliferation and cell migration in cancer[14] and pulmonary hypertension[15]. Hence, exploring regulators of HIF-1α activity will deepen our understanding of basic biological processes and offer future therapeutic strategies.

Here, we identify a molecular mechanism, where RASSF1A acts a crucial regulator of HIF-1α signaling. Upon hypoxia, RASSF1A protein is initially stabilized by reactive oxygen species (ROS)-driven and protein kinase C (PKC)-mediated phosphorylation, and is subsequently transcriptionally upregulated by HIF-1α. Vice-versa, RASSF1A directly interacts with HIF-1α, leading to increased HIF-1α stabilization, nuclear entry and transactivation of HIF-1 target genes (pyruvate dehydrogenase kinase 1 [PDK1], hexokinase 2 [HK2], and lactate dehydrogenase [LDHA]). This hitherto unknown feed-forward loop between RASSF1A and HIF-1α promotes the glycolytic shift in hypoxia-exposed human primary cells. Moreover, we provide genetic and clinical ex-vivo evidence for the function of this RASSF1A-HIF-1α loop in human lung cancer and pulmonary hypertension.

## Results

**RASSF1A is upregulated in hypoxia exposed primary lung cells.** To uncover the role of RASSF1A under physiological conditions such as hypoxia, we exposed various primary human cells to hypoxia (1% $O_2$). We observed a strong basal expression of RASSF1A mRNA in different primary human cells, namely, human broncho-alveolar epithelial cells (HBECs), pulmonary arterial-smooth muscle cells (PASMCs), -adventitial fibroblasts (PAAFs) and –endothelial cells (PAECs) as compared to A549 cell line (where RASSF1A expression is strongly reduced due to promoter hyper-methylation) (Fig. 1a). Interestingly, in all the primary human cells, RASSF1A mRNA was further increased after 24 h hypoxia exposure (Fig. 1b). In order to delineate time-dependent regulation of RASSF1A under hypoxia, we exposed PASMCs and PAAFs to hypoxia and followed its levels (Fig. 1c). Acute hypoxia exposure (15 min –6 h) strongly induced RASSF1A expression at protein level with no effect on the mRNA expression (Fig. 1d–f). Interestingly at 12 h and 24 h hypoxia exposure, both RASSF1A mRNA and protein expression were significantly upregulated (Fig. 1g–i). No increase was observed in mRNA expression of RASSF1C, another RASSF1 isoform (Supplementary Fig. 1a). Similar to PASMCs, RASSF1A was increased in PAAFs exposed to different durations of hypoxia (Supplementary Fig. 1b–d). Collectively, these data outline RASSF1A as a hypoxia-regulated protein in various primary human cells.

**Acute hypoxia enhances protein stability of RASSF1A.** To identify upstream signaling events leading to the rapid hypoxia-mediated RASSF1A protein increase, we took a genetic (si-HIF-1α) and a pharmacological approach (N-acetylcysteine: NAC) to inhibit HIF signaling and ROS formation, respectively, both known to be enhanced in hypoxia[16]. Genetic ablation of HIF-1α (Supplementary Fig. 2a) in PASMCs exposed to 15 min hypoxia had no effect on the expression of RASSF1A, while, treatment of these cells with the non-specific ROS inhibitor NAC inhibited the RASSF1A protein accumulation induced by acute hypoxia (Fig. 2a). To further specify the ROS source associated with increased RASSF1A levels, we treated human PASMCs with different inhibitors of flavoproteins and of the individual respiratory chain complexes. Hypoxia-augmented RASSF1A expression was inhibited by DPI, an inhibitor of flavoproteins and by GTK137831, a specific NADPH oxidase (NOX) 1/4 inhibitor (Supplementary Fig. 2b, c). siRNAs targeting NOX1 (Fig. 2b, Supplementary Fig. 2d), but not NOX4 (Supplementary Fig. 2e), significantly inhibited acute hypoxia-induced RASSF1A protein increase. The inhibitors of mitochondrial respiratory chain complex I (rotenone), complex II (TTFA, 3-NPA), complex III (antimycin A) and complex IV (sodium azide, $NaN_3$) did not show any effect on RASSF1A protein levels (Supplementary Fig. 2f–j). This analysis suggested that ROS production augments the protein levels of RASSF1A under acute hypoxic conditions, with a particular role of NOX1.

ROS have been reported to directly and indirectly activate PKC and ATM (Ataxia telangiectasia mutated) kinase[17] and both kinases have been implicated in the phosphorylation of RASSF1A and regulation of its function[18,19]. To test the role of these kinases on RASSF1A phosphorylation upon hypoxia, we treated human PASMCs with ATM kinase inhibitor (Ku55933) or PKC inhibitor (Gö6976) and exposed them to 15 min hypoxia. The PKC inhibitor, not the ATM kinase inhibitor, attenuated RASSF1A expression (Fig. 2c and Supplementary Fig. 3a). To directly address the phosphorylation of RASSF1A under short hypoxia exposure, we immunoprecipitated phosphoserine modified proteins from HEK293 cells overexpressing RASSF1A under hypoxia in the presence/absence of Gö6976. Immunoblotting for RASSF1A showed a two-fold increase in its phosphorylation under hypoxia, which was completely abolished by treatment with Gö6976 (Fig. 2d, e). Furthermore, using specific siRNAs, targeting PKC-A and PKC-B isoforms, we found that the increase in RASSF1A expression in hypoxia requires PKC-A, while PKC-B

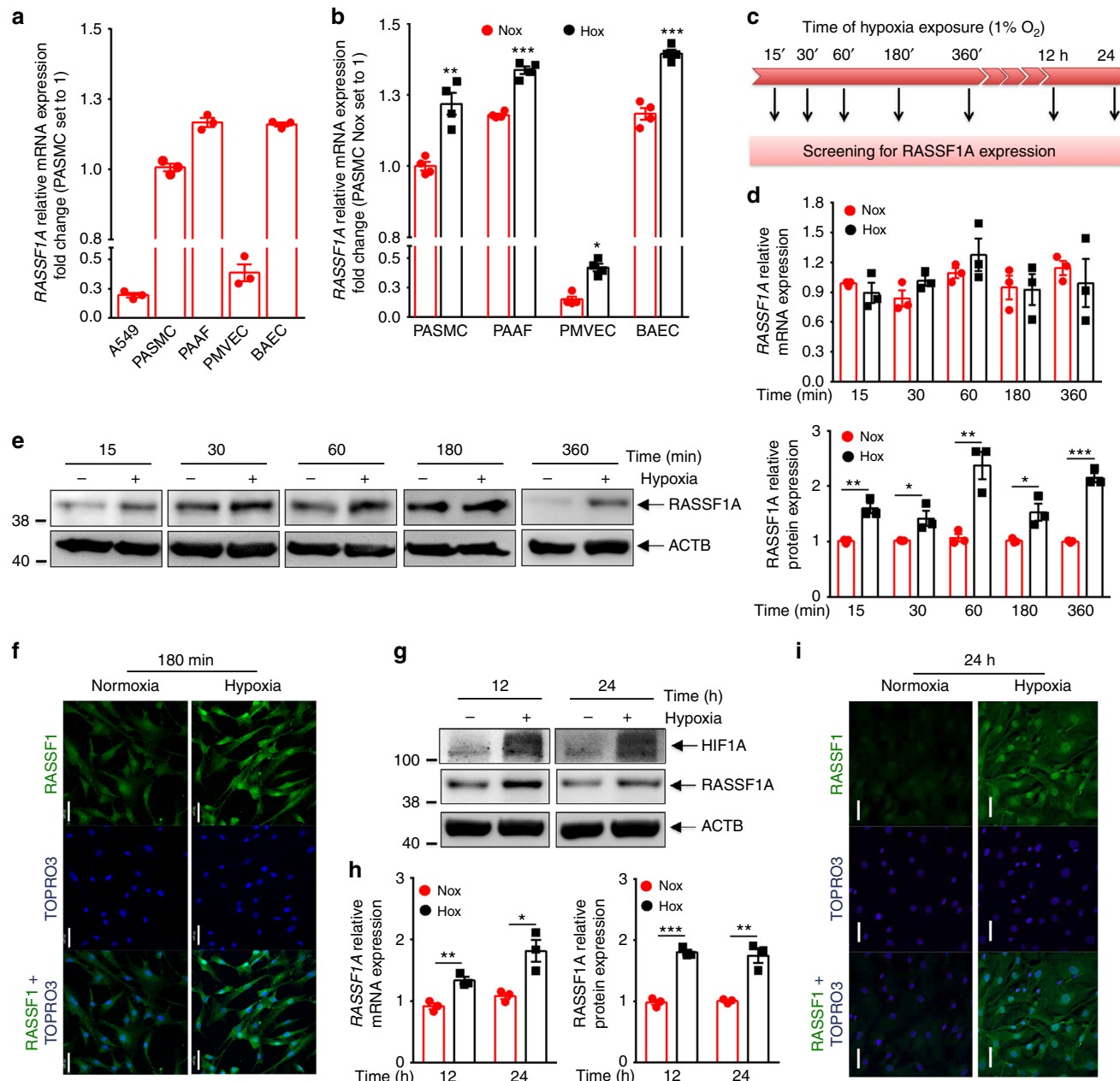

**Fig. 1** RASSF1A is upregulated in hypoxia exposed primary lung cells. **a** Relative mRNA expression of RASSF1A in human primary cells: bronchial airway epithelial cells (HBECs), pulmonary arterial-smooth muscle cells (PASMCs), -adventitial fibroblasts (PAAFs), pulmonary microvascular endothelial cells (PMVECs) and A549 (lung carcinoma cell line). **b** BAECs, PASMCs, PAECs, and PAAFs were exposed to 21% $O_2$ (normoxia: Nox) or 1% $O_2$ (hypoxia: Hox) for 24 h, followed by screening for RASSF1A mRNA expression. **c** Scheme for screening for RASSF1A expression. **d, e, g, h** Human PASMCs were exposed to normoxia hypoxia for indicated intervals. Cell lysates from each time point were subjected to **d, h** real time PCRs and **e left–g upper**, western blotting for RASSF1A, followed by **e right–g lower**, densitometric quantification of relative RASSF1A expression. ACTB (β actin) was taken as the loading control. **f, i** Localization of RASSF1A was detected by immunostaining with RASSF1 monoclonal antibody in human PASMCs at indicated hypoxia intervals. Scale bar: 50 μm. *$P < 0.05$, **$P < 0.01$, ***$P < 0.001$ compared to normoxia, unpaired Student's $t$-test. $n = 3$ independent experiments from 3 biological replicates

inhibition had no effect (Fig. 2f and Supplementary Fig. 3b, c). To further investigate the role of phosphorylation on RASSF1A levels, we generated a phospho-mimic S203D-RASSF1A and a non-phosphorylable S203A-RASSF1A mutant. S203D RASSF mutant levels were markedly increased compared to wild type (WT) RASSF1A under normoxia, and showed a further increase upon hypoxia, while S203A-RASSF1A mutant showed reduced expression compared to WT in hypoxia (Supplementary Fig. 3d). To study the effect of RASSF1A phosphorylation on its stability

over time, we treated HEK cells that overexpress RASSF1A WT, S203A or S203D mutant with CHX (30 μg/ml) for 1 and 3 h, followed by cell lysis and screening for RASSF1A expression. We observed that turnover of RASSF1A-S203A mutant was more rapid than that of RASSF1A WT while the S203D mutant showed much decreased turnover (Fig. 2g). Furthermore, to prove that the higher turnover of S203A mutant was dependent on proteasome-mediated degradation, HEK cells overexpressing RASSF1A-S203A mutant were pretreated with the proteasome

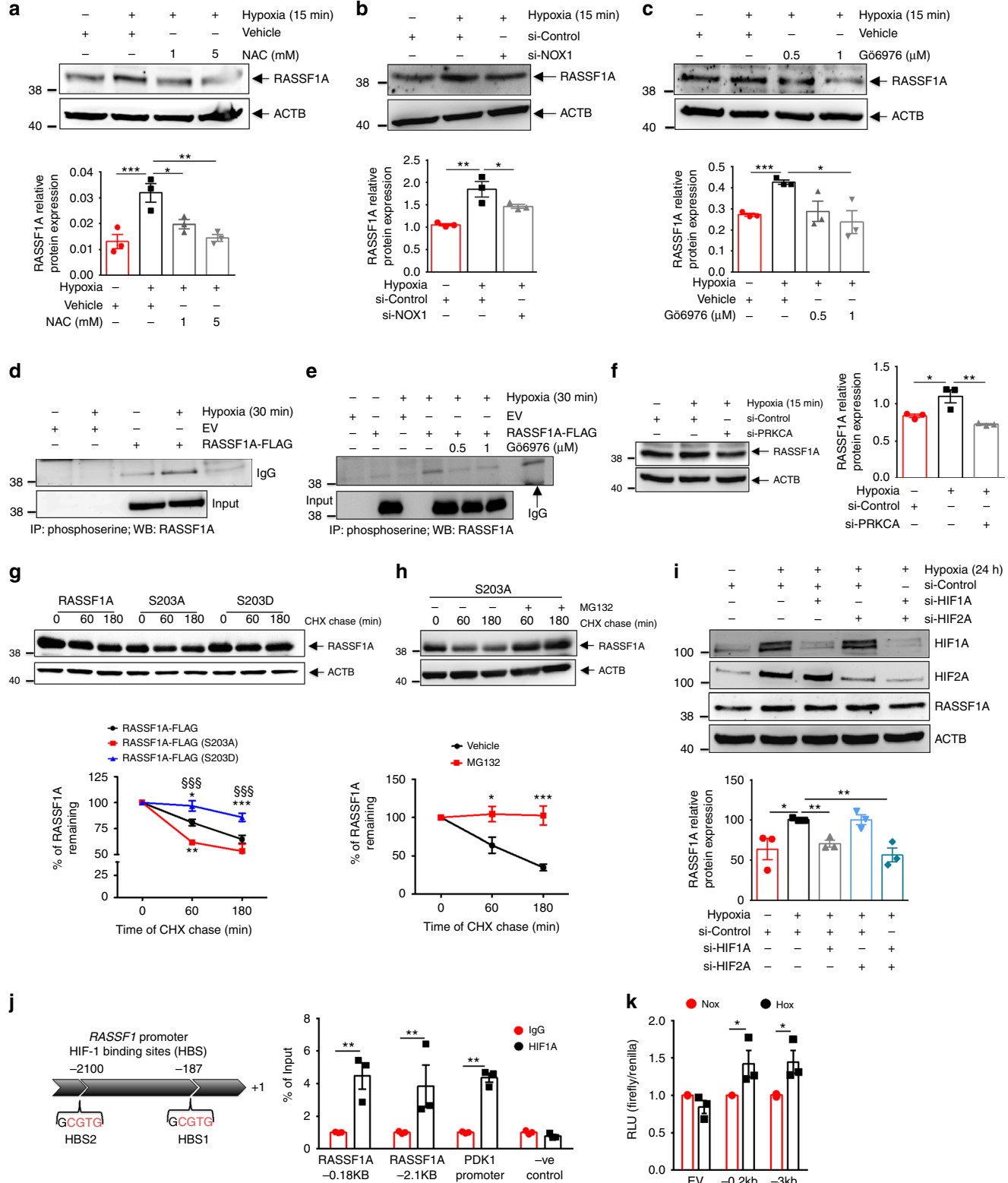

inhibitor MG132, followed by CHX chase experiment. Treatment with MG132 successfully reversed the increased turnover of RASSF1A-S203A (Fig. 2h).

These data suggest that PKC-A phosphorylates and stabilizes RASSF1A upon hypoxia. PKC-A activation and RASSF1A phosphorylation and stabilization are induced by NOX1 activation and elevated ROS formation upon hypoxia.

**HIF-1α regulates RASSF1A at transcriptional level.** Interestingly, at longer time points of hypoxia (12–24 h), an increase in RASSF1A mRNA expression was observed. This was similarly noted when human PASMCs were treated with the PHD inhibitor dimethyloxalylglycine (DMOG), which stabilizes HIF (Supplementary Fig. 4a, b). These observations indicate that RASSF1A expression might additionally be transcriptionally

**Fig. 2** Hypoxia promotes RASSF1A protein stability and HIF1α regulates its transcription. **a** Human PASMCs were pre-treated with NAC (N-acetyl cysteine) at indicated concentrations for 1 h or **b** transfected with NOX1 siRNA for 48 h or **c** pre-treated with Gö6976 at the indicated concentrations for 1 h. **a**–**c** Treatments were followed by 15 min hypoxia exposure, **a**–**c**, **upper** western blotting and **a**–**c lower** quantification of RASSF1A expression. **d**, **e** HEK 293 cells were transfected with RASSF1A-FLAG or empty vector (EV), and exposed to hypoxia for 30 min, **d** without or **e** with pre-treatment of Gö6976 for 1 h, followed by phosphoserine IP and RASSF1A blotting. **f** Human PASMCs were transfected with PKCα siRNA (si-PRKCA) for 48 h, followed by 15 min hypoxia exposure, western blotting, and **f**, **right** densitometric quantification of RASSF1A. **g**, **h** HEK293 cells were transfected with plasmids as indicated, treated with 30 μg/ml cycloheximide (CHX) without **g** or with **h** MG132 pretreatment, followed by hypoxia exposure for 1 h and 3 h and western blotting. **g**, **h**, **lower** Densitometrical quantified data of % of RASSF1A remaining. **i** Human PASMCs were transfected with HIF1α siRNA (si-HIF1A) or HIF2α siRNA (si-HIF2A) or both, followed by 24 h hypoxia exposure, western blotting for indicated proteins **i**, **upper** and densitometric quantification **i**, **lower** of RASSF1A. **j**, **left** In silico analysis of HIF-1 binding sites (HBS) in human *RASSF1A* promoter. **j**, **right** Human PASMCs were exposed to hypoxia for 24 h, followed by ChIP with anti-HIF1α (HIF1A) and real time PCR with primers spanning the putative HBS sites in *RASSF1A* promoter. **k** HEK293 cells were transfected with indicated luciferase promoter plasmids, followed by 24 h hypoxia exposure and luciferase activity measurement. RLU relative luciferase units. *$P < 0.05$, **$P < 0.01$, ***$P < 0.001$ compared to **b**, **f**, **i** si-Control (Hypoxia), **a**, **c**, **h** Vehicle, **g** RASSF1A-FLAG at same time point, **j** IgG or **k** EV, one-way ANOVA followed by SNK multiple comparison test. **g** §§§$P < 0.001$ compared to RASSF1A-FLAG (S203A), two-way ANOVA. Data represent mean ± s.e.m. $n = 3$ independent experiments from 3 biological replicates for human PASMCs and $n = 3$ independent experiments for HEK cells

regulated by HIF transcription factors (HIF-1α and HIF-2α). To examine this possibility, we studied the impact of siRNA-mediated knockdown of HIF-1α and HIF-2α on RASSF1A expression under conditions of 24 h hypoxia. si-HIF-1α significantly reduced and near normalized RASSF1A expression compared to si-Control (hypoxia) at mRNA (Supplementary Fig. 4c) as well as protein level (Fig. 2i), while si-HIF-2α did not exert any effect. To further examine whether HIF-1α directly binds to the RASSF1A promoter, we analyzed the human RASSF1 gene promoter sequence for HIF binding sites (HBS) and found two candidate sites at −187 bp and −2.1 kb (Fig. 2j left panel). To determine whether HIF-1α directly binds at these sites in the RASSF1A promoter, we performed ChIP assays in human PASMCs exposed to hypoxia for 24 h. These assays revealed a strong enrichment of HIF-1α binding in cells exposed to hypoxia (Fig. 2j right panel). Further, to test whether the HBS in the RASSF1 promoter function as HRE, we inserted 200-bp (containing HBS1) and 3000-bp (containing HBS1 and HBS2) fragments encompassing one or both binding sites upstream of firefly luciferase coding sequences in the pGL3 plasmid. HEK293 cells were transfected with the RASSF1A-HBS1 or -HBS1 plus HBS2 reporter and a control reporter (pGL3-Renilla) and exposed to hypoxia for 24 h. Both RASSF1A-HBS1 and -HBS1 plus HBS2 significantly increased luciferase activity in hypoxic HEK293 cells (Fig. 2k). Collectively, these data indicate that increased expression of RASSF1A under sustained hypoxia is mediated by HIF-1 binding to sites in the RASSF1 promoter.

**RASSF1A promotes hypoxic proliferation and glycolytic shift.** A large body of evidence indicates that hypoxic challenge of pulmonary vascular cells leads to an altered vascular phenotype (i.e., increased proliferation, survival and metabolic reprogramming to aerobic glycolysis), with subsequent pulmonary vascular remodeling[20], leading to development of hypoxic PH. To test whether RASSF1A has a causative role in the induction of this phenotype, we carried out knockdown and overexpression of RASSF1A in both human PASMCs and PAAFs exposed to hypoxia, followed by assessment of proliferation, and expression of metabolic genes. The increase in proliferation of human PASMCs and PAAFs in response to hypoxia was significantly decreased by siRNA-mediated knockdown of RASSF1 (si-RASSF1; Fig. 3a, Supplementary Fig. 5a), while RASSF1A overexpression (RASSF1A-FLAG) further boosted the hypoxia-induced proliferation (Fig. 3b, Supplementary Fig. 5b). Importantly, si-RASSF1 was similarly potent as si-HIF-1α, to block the hypoxia-induced proliferative response, whereas the combination of si-RASSF1 and si-HIF-1α did not exert additional anti-proliferative effects (Fig. 3c).

Similar to HIF-1α inactivation, si-RASSF1 significantly decreased mRNA expression of PDK1, LDHA, and HK2 in hypoxic human PASMCs and PAAFs (Fig. 3d and Supplementary Fig. 5c). Correspondingly, protein levels of PDK1, LDHA, and HK2 were also reduced by si-RASSF1 treatment (Fig. 3e). Conversely, RASSF1A-FLAG led to a further increase in mRNA (Fig. 3f and Supplementary Fig. 5d) and protein (Fig. 3g) levels of PDK1, LDHA, and HK2 in hypoxic human PASMCs. To further substantiate the functional effect of RASSF1A on glycolysis, lactate, the end product of glycolysis, was measured in cells transfected with empty vector or RASSF1A-FLAG and exposed to hypoxia for 24 h. Increased lactate production was observed in hypoxic cells and was further augmented by RASSF1A-FLAG (Fig. 3h). Collectively, these data indicate that RASSF1A induces a metabolic switch and drives the hyper-proliferative response in hypoxia-exposed human lung vascular cells.

**RASSF1A regulates HIF-1α stability and transactivation.** We hypothesized that the hitherto unrecognized role of RASSF1A in hypoxia sensing and signaling could be executed through modulation of HIF-1α activity. To evaluate this hypothesis, we inactivated RASSF1 in human PASMCs and PAAFs with siRNA. This resulted in a significant decrease in HIF-1α protein by 70% and 95% in hypoxic human PASMCs and PAAFs, respectively (Fig. 4a, Supplementary Fig. 6a). Further, RASSF1A-FLAG overexpression, in both human PASMCs and PAAFs led to a further increase in HIF-1α protein (Fig. 4b, Supplementary Fig. 6b). Interestingly, only RASSF1A overexpression but not RASSF1C, was able to rescue the effect of RASSF1 knockdown on HIF-1α protein expression and LDHA/PDK1 expression (Supplementary Fig. 6c, d). To determine whether RASSF1A influences HIF-1 mediated transcriptional activity, HeLa cells were co-transfected with the HIF-1-dependent reporter plasmid HRE-pGL3, containing 5 repeats of an HRE upstream of firefly luciferase, and control reporter pGL3-Renilla; in presence of si-RASSF1/si-control or RASSF1A-FLAG/empty vector. Transfected cells were exposed to hypoxia for 24 h. si-RASSF1 significantly reduced HIF-1 transcriptional activity as observed by a decrease in the firefly luciferase activity (Fig. 4c). Conversely, RASSF1A-FLAG increased the luciferase activity 1.8-fold (Fig. 4c), while, RASSF1C overexpression did not have an effect on HIF1A protein expression and its transcriptional activity (Supplementary Fig. 6e, f). Accordingly, RASSF1A-FLAG markedly increased HIF-1α occupancy at the PDK1, LDHA, and HK2 gene HREs as assessed by ChIP analysis (Fig. 4d).

Further, to study the effect of RASSF1A phosphorylation on HIF-1α stability and function, we carried out overexpression of RASSF1A WT, S203A or S203D mutant in human PASMCs,

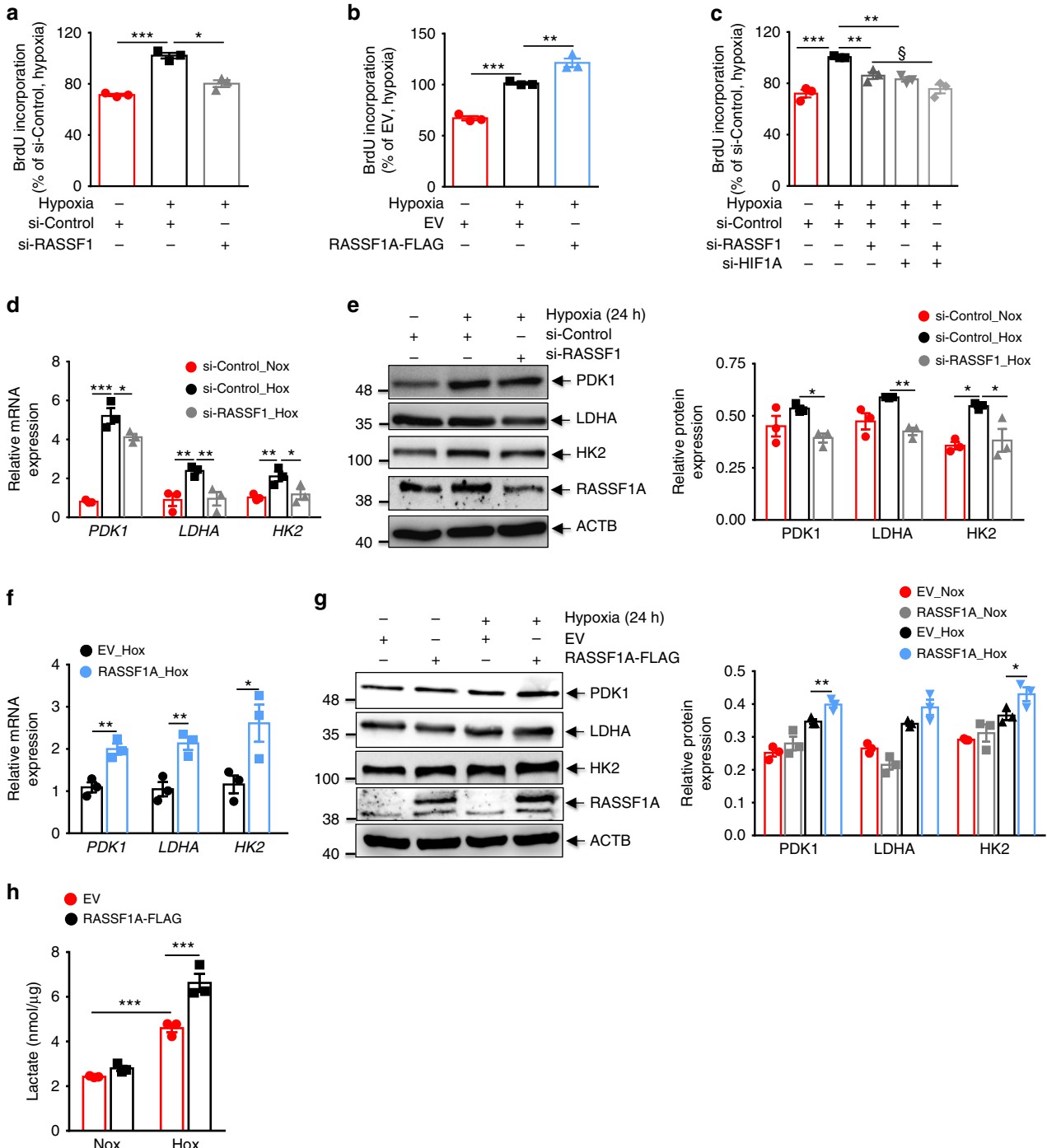

**Fig. 3** RASSF1A promotes proliferation and glycolytic shift in hypoxic human PASMCs. Human PASMCs (**a–c**) were transfected with **a** RASSF1 siRNA (si-RASSF1) and control siRNA (si-Control) or **c** si-RASSF1 and HIF1α siRNA (si-HIF1A) in combination or **b** RASSF1A-FLAG and empty vector (EV). 24 h after transfection, cells were exposed to normoxia or hypoxia for 48 h and proliferation was measured by BrdU incorporation assay. Human PASMCs were transfected with **d, e** si-RASSF1 and si-Control or **f, g** RASSF1A-FLAG and EV. 24 h after transfection, cells were exposed to normoxia or hypoxia for 24 h. **d, f** Real time PCRs for indicated genes were performed. **e, g** Cell lysates were subjected to (**left**) western blotting, followed by (**right**) densitometric quantification of relative PDK1, LDHA and HK2 to ACTB expression. **h** HEK293 cells were transfected with RASSF1A-FLAG or EV and 24 h after transfection, exposed to normoxia or hypoxia for 24 h and intracellular lactate production was measured. *$P < 0.05$, **$P < 0.01$, ***$P < 0.001$ compared to **a, c–e** si-Control (hypoxia) or **b, f–h** EV (hypoxia), one-way ANOVA followed by SNK multiple comparison test. **c** §$P < 0.01$ compared to si-RASSF1, two-way ANOVA. $n = 3$ independent experiments from 3 biological replicates each; data represent mean ± s.e.m.

followed by hypoxia exposure for 24 h. Overexpression of RASSF1A-S203D mutant increased expression of HIF-1α, similar to RASSF1A WT, while S203A mutant failed to show this effect (Fig. 4e). Consequently, S203A mutant did not lead to an increase

in HIF-1α transcriptional activity as measured by HRE luciferase assay (Fig. 4f) in HeLa cells.

RASSF1A is a well-documented regulator of hippo signaling and Yap transcriptional activity (Supplementary Fig. 7a) and therefore,

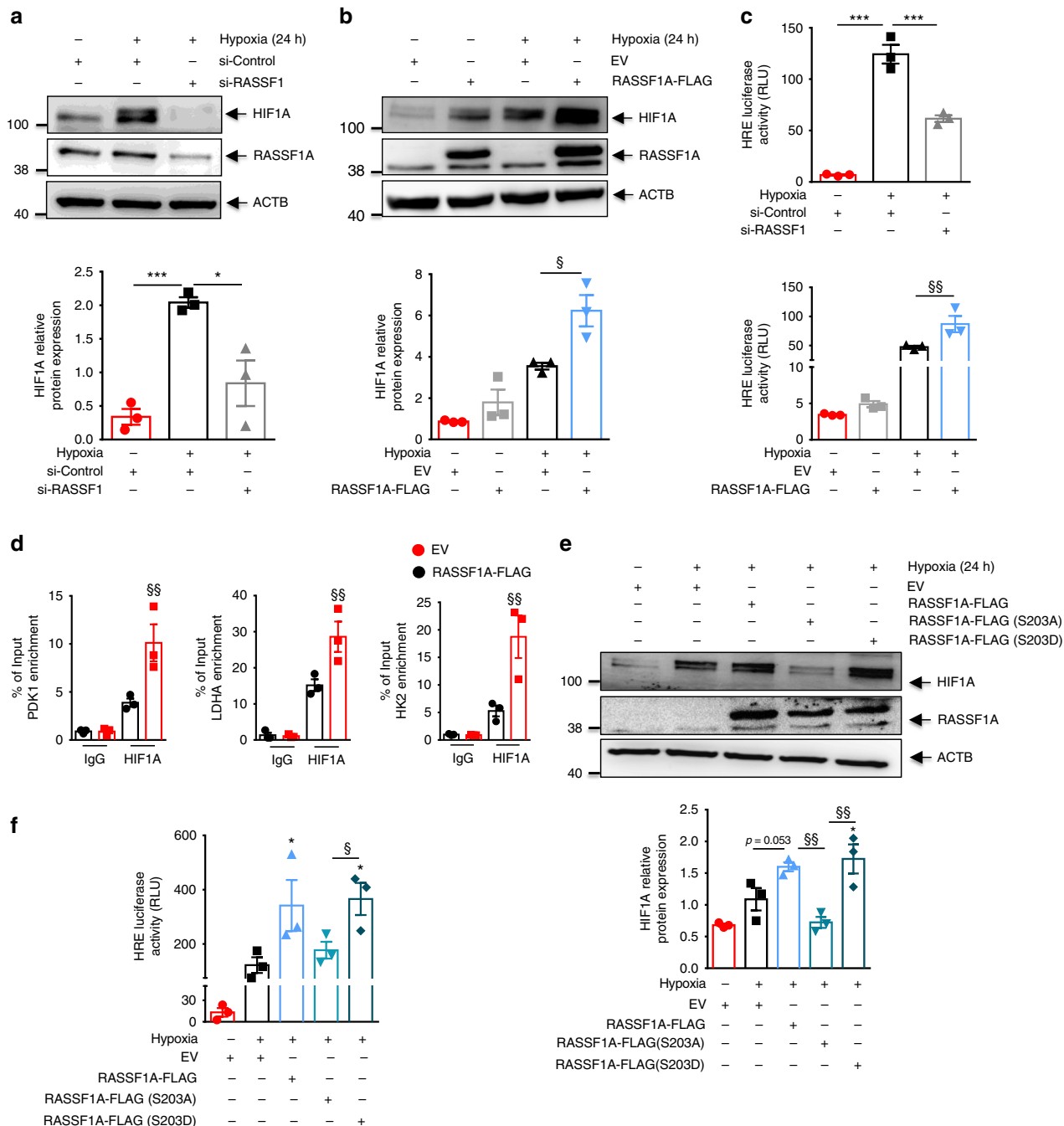

**Fig. 4** RASSF1A regulates HIF1α protein stability and transcriptional activity. **a**, **b** Human PASMCs were transfected with **a** RASSF1 siRNA (si-RASSF1) and control siRNA (si-Control) or **b** RASSF1A-FLAG plasmid or EV. 24 h after transfection, cells were exposed to hypoxia or normoxia for further 24 h. Cell lysates were subjected to **a**, **b**, **upper** western blotting for indicated proteins, followed by **a**, **b**, **lower** densitometric quantification of relative RASSF1A expression. **c** A luciferase reporter under control of multiple HIF1α binding sites was transfected into HeLa cells with **c**, **upper** si-RASSF1 or **c**, **lower** RASSF1A-FLAG. 6 h after transfection, cells were exposed to hypoxia for 24 h. Cells were lysed and luciferase activity was measured and normalized to co-transfected Renilla luciferase internal control. RLU relative luciferase units. **d** From hypoxic HEK293 cells transfected with RASSF1A-FLAG or EV, chromatin was precipitated with anti-HIF1α (HIF1A) antibody or rabbit IgG (IgG) and was analyzed by real time PCR with primers spanning the hypoxia-response element (HRE) regions of mentioned genes (PDK1, LDHA, HK2). **e** Human PASMCs were transfected with EV, RASSF1A-FLAG, RASSF1A-FLAG (S203A) or RASSF1A-FLAG (S203D). 24 h later, cells were exposed to hypoxia or normoxia for further 24 h. Cell lysates were subjected to **e**, **upper** western blotting for indicated proteins, followed by **e**, **lower** densitometric quantification of relative HIF1A expression. **f** A luciferase reporter under control of multiple HIF1α binding sites (HRE) was transfected into HeLa cells with EV, RASSF1A-FLAG, RASSF1A-FLAG (S203A) or RASSF1A-FLAG (S203D). 6 h after transfection, cells were exposed to hypoxia for 24 h. Cells were lysed and luciferase activity was measured and normalized to co-transfected Renilla luciferase internal control. $*P < 0.05$, $**P < 0.01$, $***P < 0.001$ compared to **a**, **c** si-Control (hypoxia) or **b**, **c** EV (normoxia) or **e**, **f** EV (hypoxia), $^{§}P < 0.05$, $^{§§}P < 0.01$, $^{§§§}P < 0.001$ compared to **b**–**d** EV (Hypoxia) or **e**, **f** RASSF1A-FLAG (S203A), one-way ANOVA followed by SNK multiple comparison test. Data represent mean ± s.e.m. $n = 3$ independent experiments from 3 biological replicates

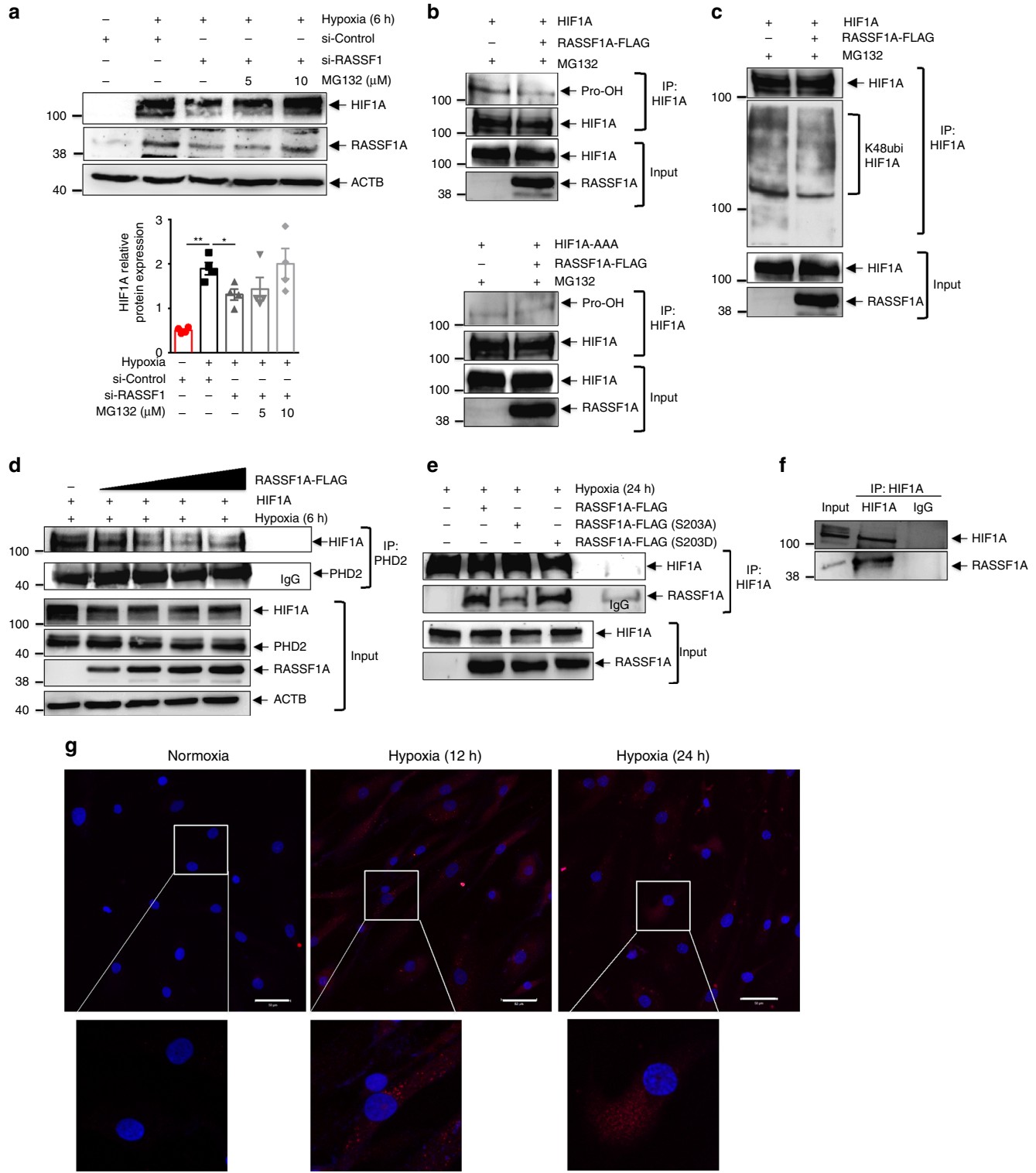

to study whether RASSF1A mediated effect on HIF-1α is hippo signaling driven, we overexpressed RASSF1A in presence of YAP wild type/mutant overexpression, followed by hypoxia exposure. Both YAP wild type/mutant overexpression further enhanced the hypoxia-driven stabilization of HIF-1α. However, it did not have any further effect on increased HIF-1α stabilization observed under RASSF1A overexpression (Supplementary Fig. 7b). This finding was further substantiated by HRE luciferase assay where again YAP overexpression (wild type or mutant) did not show any additional effect on RASSF1A mediated HIF-1α transcriptional

activity (Supplementary Fig. 7c). Taken together, these data imply a role of RASSF1A in the regulation of HIF-1α expression and its transcriptional activity, independent of hippo signaling.

As RASSF1A overexpression leads to an increase in HIF-1α at the protein level but not at the mRNA level (Supplementary Fig. 7d), we hypothesized that RASSF1A mediates its effect by increasing HIF-1α stability. Thus, we investigated whether RASSF1A stabilizes HIF-1α by inhibiting its prolyl hydroxylation, its ubiquitination by von Hippel-Lindau protein (pVHL) and subsequent proteasomal degradation. We transfected human

**Fig. 5** RASSF1A interacts with HIF1α. **a** Human PASMCs were transfected with si-RASSF1 and 48 h after transfection, pre-treated with indicated concentrations of MG132 for 30 min, followed by 6 h hypoxia exposure. Cell lysates were subjected to **a**, **upper** western blotting for indicated proteins, followed by **a**, **lower** densitometric quantification of relative HIF1A expression. *$P < 0.05$ compared to si-Control (hypoxia), one-way ANOVA followed by SNK multiple comparison test. Data represent mean ± s.e.m. $n = 3$ independent experiments from 3 biological replicates. **b**, **c** HEK293 cells were transfected with plasmids indicated on top of lanes. 24 h after transfection, cells were treated with 25 μM MG132, followed by 5 h hypoxia exposure. HIF1α (HIF1A) was immunoprecipitated (IP), followed by western blotting for **b** hydroxyl proline (Pro-OH) or **c** lys48 ubiquitin (K48ubi) antibody. **d** HEK293 cells were transfected with plasmids indicated on top of lanes and exposed to hypoxia for 6 h, followed by PHD2 IP and western blotting for indicated proteins. **e** HEK293 cells were transfected with plasmids indicated on top of lanes and exposed to hypoxia for 24 h. HIF1A IP and RASSF1A co-IP were detected by western blotting. **f** Human PASMCs were exposed to hypoxia for 24 h followed by **f** HIF1A IP and RASSF1A western blotting and **g** proximity ligation assay with HIF1A and RASSF1A antibodies. Each red spot represents for a single interaction between HIF1A and RASSF1A and DNA was stained with DAPI (blue). Scale bar: 50 μm. $n = 2$–3 independent experiments

PASMCs with si-RASSF1, followed by treatment with different concentrations of the proteasome inhibitor MG132 under hypoxia. Notably, the decrease observed in HIF-1α protein levels after knockdown of RASSF1A was reduced in a dose-dependent manner by MG132 treatment (Fig. 5a). To further confirm this observation, HIF-1α was overexpressed in HEK cells in the presence of empty vector or RASSF1A-FLAG. 24 h after transfection, cells were exposed to hypoxia in the presence of 25 μM MG132 for 5 h. Cells lysates were immunoprecipitated with anti-HIF-1α antibody and immunoblotted with anti-ubiquitin (K48ubi) and anti-hydroxyproline (Pro-OH) antibodies. RASSF1A-FLAG expression led to decreased prolyl hydroxylation and concomitantly decreased ubiquitination of HIF-1α (Fig. 5b upper panel, c). In contrast, no prolyl hydroxylation was observed for the HIF-1α AA mutant (P402A/P564A) in presence or absence of RASSF1A-FLAG expression (Fig. 5b lower panel) indicating that the effect of RASSF1A on HIF-1α stabilization is indeed executed via inhibition of its prolyl hydroxylation. Moreover, co-immunoprecipitation (Co-IP) assays using anti-HIF-1α and anti-PHD2 antibodies demonstrated that RASSF1A-FLAG dose-dependently decreased the interaction of HIF-1α with PHD2 in hypoxic cells (Fig. 5d).

RASSF1A serves as a scaffold for the assembly of multiple protein complexes, thereby regulating various cellular functions[5]. We thus hypothesized that a physical interaction of RASSF1A with HIF-1α may underlie its impact on hypoxic signaling. To test this hypothesis, we performed Co-IP assays in hypoxic cells expressing RASSF1A-FLAG and HIF-1α. RASSF1A-FLAG was immunoprecipitated by anti-HIF-1α antibody and vice versa (Fig. 5e, Supplementary Fig. 8a, b). Interestingly, as compared to RASSF1A WT and S203D mutant, S203A mutant displayed decreased interaction with HIF-1α in the pull down experiments. (Fig. 5e). Lack of RASSF1C-HIF1α interaction was also obvious upon overexpression of RASSF1C-FLAG (Supplementary Fig. 8c). Similarly, endogenous RASSF1A was immunoprecipitated specifically by anti-HIF-1α antibody, but not by IgG (Fig. 5f) in human PASMCs exposed to hypoxia. We were further able to substantiate these results by performing proximity ligation assay (PLA), where we observed that RASSF1A-HIF-1α interaction was majorly confined to the cytosolic compartment at different time points of hypoxia (Fig. 5g), in agreement with predominant cytosolic localization of RASSF1A (Supplementary Fig. 8d). Taken together, these findings strongly suggest that RASSF1A interacts with HIF-1α, preventing its prolyl hydroxylation, ubiquitination and degradation and hence, increasing its transcriptional activity.

**RASSF1A is increased in pulmonary vessels of PH patients.** An excellent model system for studying the role of hypoxia in disease pathogenesis is PH, where chronic exposure to alveolar hypoxia results in enhanced proliferation of smooth muscle cells and adventitial fibroblasts of small pulmonary arteries, leading to vessel wall thickening and luminal occlusion[21]. Indeed, chronic hypoxia plays a key pathogenic role in PH associated with chronic obstructive pulmonary disease (COPD), interstitial lung diseases, sleep-disordered breathing and chronic exposure to high altitude, and hypoxia-induced PH is regarded as a distinct diagnostic entity by the World Health Organization (WHO class III)[22,23]. In order to determine the expression pattern of RASSF1A in human healthy donor and PH patient (Idiopathic pulmonary arterial hypertension/IPAH, COPD-PH) lungs, we carried out laser-assisted micro-dissection of pulmonary vessels, followed by RNA isolation and real time PCRs with gene specific primers. Interestingly, we found significantly elevated RASSF1A mRNA in vessels from IPAH (Fig. 6a upper panel) and COPD-PH (Fig. 6a lower panel) samples as compared to healthy donor tissue. We additionally observed increased expression of HIF-1 regulated genes, LDHA and HK2, in both IPAH and COPD-PH vessels (Fig. 6a). In agreement with these findings, protein levels of RASSF1A, LDHA and PDK1 were increased in pulmonary arteries of IPAH patients in comparison to healthy donors (Supplementary Fig. 9a). Immunofluorescence staining of human lungs (IPAH, COPD-PH, and donors) demonstrated strong immunoreactivity of RASSF1A in the pulmonary arteries of IPAH and COPD-PH patients as compared to donors (Fig. 6b). Interestingly, co-immunostaining with HIF-1α antibody revealed co-localization of RASSF1A and HIF-1α in the pulmonary vessels (Fig. 6c). Thus, in addition to the post-transcriptional stabilization of RASSF1A upon acute hypoxic exposure of human cells, the analysis of PH patients shows that also RASSF1A mRNA levels are increased in pulmonary vessels. This mRNA induction is accompanied by the activation of HIF-1 downstream targets in these diseased vessels.

**RASSF1A drives proliferation/glycolytic shift in IPAH PASMCs.** Hyper-proliferative and apoptosis-resistant PASMCs in IPAH patients exhibit increased expression of HIF-1α and a shift to glycolytic metabolism[20,24], similar to the response of control human PASMCs exposed to prolonged hypoxia. PASMCs from healthy donors and IPAH patients were isolated and analyzed for expression of RASSF1A. Increased expression of RASSF1A mRNA and protein was observed in IPAH-PASMCs as compared to donors (Fig. 6d, e). Similarly, increased expression was observed in IPAH-PAAFs in comparison to donors (Supplementary Fig. 9b, c). As our above studies put forward an explicit role of RASSF1A in regulating proliferation and metabolic gene expression in ex vivo hypoxia-exposed human PASMCs, we further investigated whether RASSF1A exerts a corresponding function in diseased PASMCs originating from IPAH patients. Indeed, si-RASSF1 significantly reduced the proliferation of IPAH PASMCs (Fig. 6f). Similar to the results obtained in hypoxia-exposed donor PASMCs (Fig. 3c), si-RASSF1 and si-HIF-1α decreased proliferation of IPAH PASMCs equally, whereas a combination of both did not display any additional effect (Supplementary Fig. 9d). Moreover,

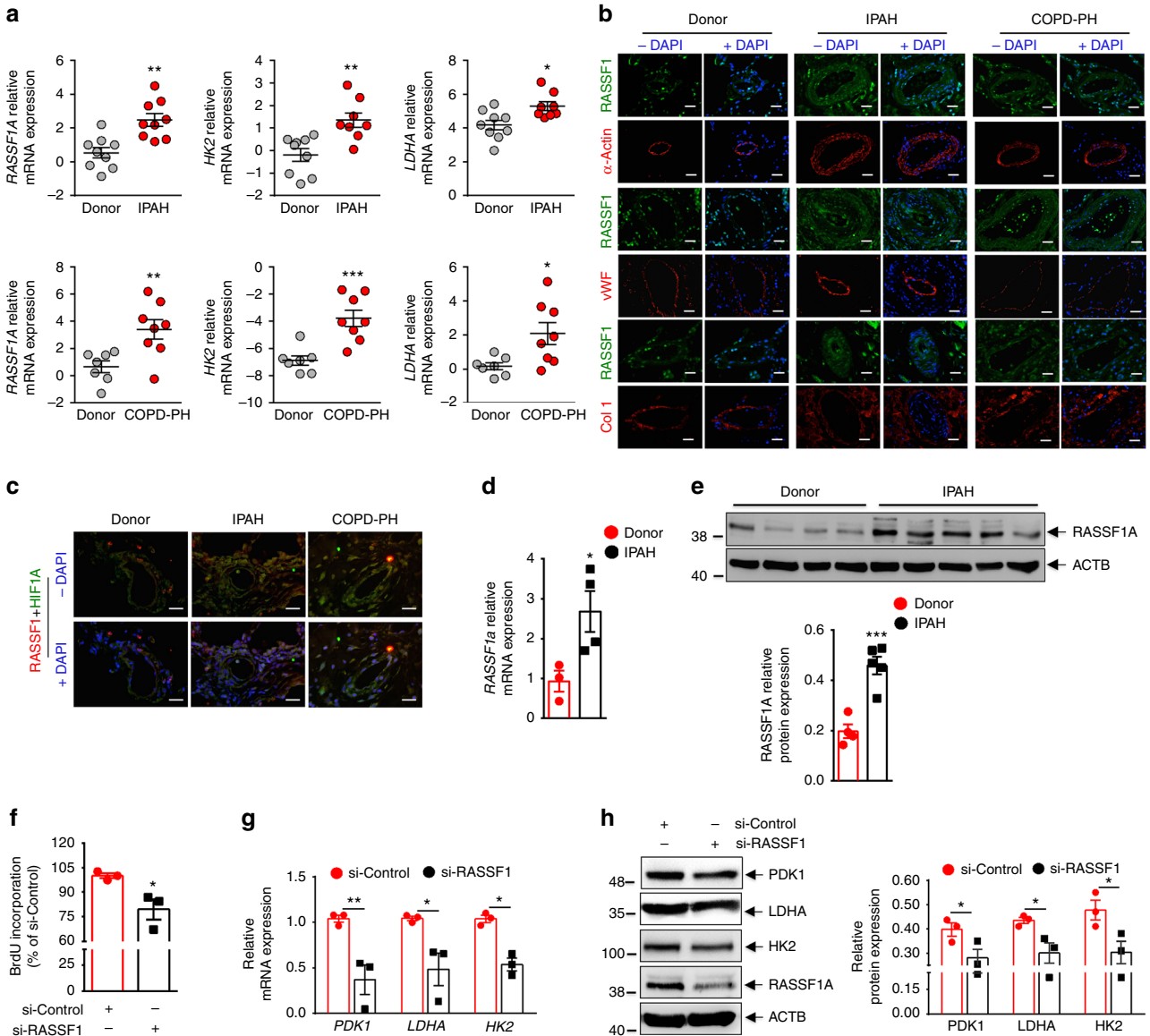

**Fig. 6** RASSF1A regulates proliferation and glycolytic metabolism in IPAH-PASMCs. **a** Pulmonary vessels from frozen lung sections of **a**, **upper** IPAH patients ($n = 9$), **a**, **lower** COPD-PH ($n = 8$) patients and donors ($n = 8$–9) were collected via laser-assisted microdissection. RNA was isolated and real time PCRs for indicated genes were performed. **b**, **c** Representative paraffin lung tissue sections from donors, IPAH patients, and COPD-PH patients were subjected to immunofluorescence staining of RASSF1, HIF1α (HIF1A), alpha smooth muscle actin (α-Actin), von-willebrand factor (vWF) and collagen 1 (Col1). Nuclei are counterstained with DAPI (blue). Scale bar: 20 μm. **d**, **e** Expression of RASSF1A in IPAH- vs donor-PASMCs as analyzed using **d** real time PCRs and **e**, **upper** western blotting, followed by **e**, **lower** densitometric quantification of relative RASSF1A expression. **f** Human PASMCs from IPAH patients were transfected with RASSF1 siRNA (si-RASSF1) and control siRNA (si-Control). 6 h after transfection, cells were placed in medium with growth factors for 48 h. Proliferation was measured by BrdU incorporation assay. **g**, **h** Cells were treated with siRNA as above-mentioned and cell lysates were subjected to **g** real time PCRs and **h**, **left** western blotting, followed by **h**, **right** densitometric quantification of relative PDK1, LDHA, and HK2 to ACTB expression. *$P < 0.05$, **$P < 0.01$, ***$P < 0.001$ compared to **a–e** donor or **f–h** si-Control, unpaired Student's $t$-test. $n = 3$ independent experiments from 3 biological replicates, Data represent mean ± s.e.m.

si-RASSF1 resulted in a strong decrease in the expression of the glycolytic enzymes PDK1, LDHA and HK2 at both the mRNA (Fig. 6g) and protein level (Fig. 6h). These results suggest that RASSF1A has a central function in both the metabolic switch and in the hyper-proliferative feature of pulmonary vascular adventitial fibroblasts and smooth muscle cells from IPAH patients.

**RASSF1A knockout mice do not develop hypoxia-induced PH.** Notably, Rassf1a expression was significantly increased in lungs of mice exposed to hypoxia compared to normoxic mice (Fig. 7a,

Supplementary Fig. 10). To explore the role of RASSF1A in promoting hypoxia-induced PH in vivo, we exposed wild type littermates (WT), Rassf1a heterozygous ($Rassf1a^{+/-}$) and homozygous ($Rassf1a^{-/-}$) knockout (KO) mice to hypoxia (10% $O_2$) or normoxia for 4 weeks. We analyzed the effects by magnetic resonance imaging (MRI) and cardiopulmonary hemodynamic measurements employing right heart catheterization. Under normoxic conditions, there were no differences in hemodynamic parameters such as right ventricular systolic pressure (RVSP) between WT and Rassf1a KO mice. In contrast to WT mice, which develop characteristic PH phenotypes after chronic hypoxia

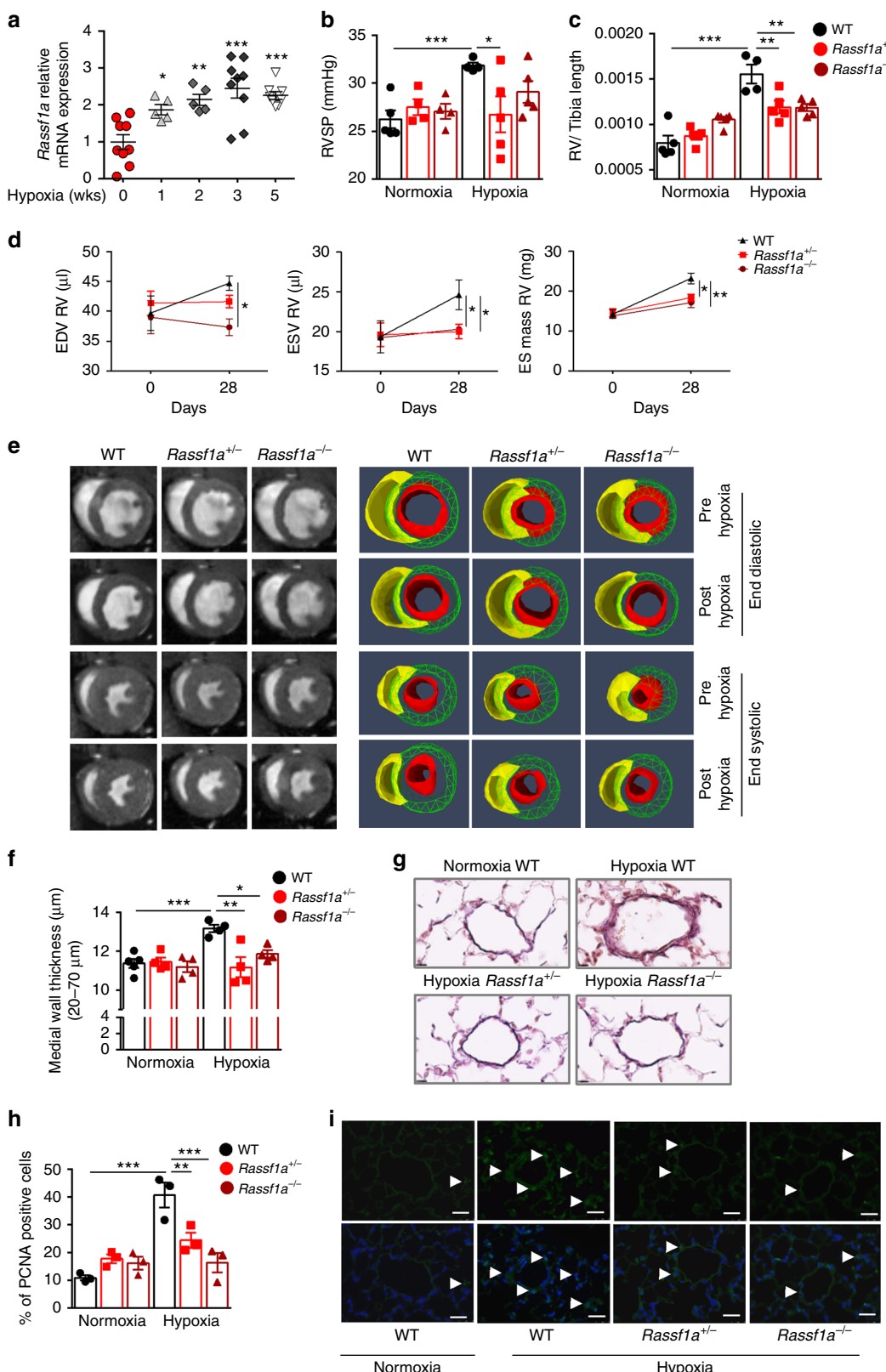

exposure, both homozygous ($Rassf1a^{-/-}$) and heterozygotes ($Rasssf1a^{+/-}$) KO mice exhibited significantly lower RVSP (Fig. 7b) and did not develop hypertrophy of the right ventricle (RV) (measured by RV/tibia length ratio; Fig. 7c). Additionally, pre-hypoxic and post-hypoxic MRI measurements revealed that Rassf1a inactivation protected mice against chronic hypoxia

evoked right heart abnormalities such as enlargement of RV, end-diastolic volume (EDV) increase, end-systolic volume (ESV) increase and augmentation of RV mass (Fig. 7d, e). In line with the observed hemodynamic changes, Rassf1a KO mice did not show an increase in the medial wall thickness of resistance pulmonary arteries (<70 μm) as seen in the hypoxic WT mice (Fig. 7f,

**Fig. 7** *Rassf1a* knockout protects mice against Hypoxia induced PH. **a** Real time PCR measurement of Rassf1a mRNA in lungs from WT mice exposed to normoxia ($n = 9$) or hypoxia for 1 week ($n = 5$), 2 weeks ($n = 5$), 3 weeks ($n = 9$), and 5 weeks ($n = 9$). **b–i** Rassf1a KO mice (*Rassf1a$^{+/-}$*, *Rassf1a$^{-/-}$*) and wildtype littermates (WT) were exposed to Hypoxia (10% $O_2$) for 28 days or maintained under normoxic conditions. **b, c** Physiological measurements were carried out on all groups of mice exposed to hypoxia and normoxia to determine **b** right ventricle systolic pressure (RVSP) and **c** RV hypertrophy (RV/ tibia length). **d** All three groups of mice exposed to hypoxia were subjected to magnetic resonance imaging (MRI) pre- and post-hypoxia to determine right heart function presented as **d**, **left** end diastolic volume (EDV), **d**, **middle**, end systolic volume (ESV) and **d**, **right** end systolic (ES) mass. **e** Representative end diastolic and systolic images for each experimental group are shown. ($n = 4$–5 mice each group). **f, g** Lung sections were subjected to H&E staining, followed by morphometric analysis of pulmonary vessels ($n = 4$–5 each group). **f** Medial wall thickness of pulmonary arteries (20- to 70-μm in diameter) and representative photomicrographs (**g**) are shown. **h, i** Immunohistochemical analysis of PCNA in small (20- to 70-μm in diameter) pulmonary vessels from the groups described above was carried out ($n = 3$ each group). **h** Percentage of PCNA positive cells and **i** representative photomicrographs for PCNA staining (green) are shown. Nuclei are counterstained with DAPI (blue). Scale bar: 20 μm. *$P < 0.05$, **$P < 0.01$, ***$P < 0.001$ compared to **a** hypoxia 0 weeks or **b–d, f, h** hypoxic WT, one-way ANOVA followed by SNK multiple comparison test. Data represent mean ± s.e.m.

g). These protective effects on lung vascular remodeling were linked to decreased cell proliferation, as assessed by proliferating cell nuclear antigen (PCNA) stainings in small pulmonary arteries (Fig. 7h, i). Collectively, these data indicate that genetic ablation of Rassf1a is protective against hypoxia-induced lung vascular remodeling and RV hypertrophy and dilatation, and that ablation of one Rassf1a allele (*Rasssf1a$^{+/-}$*) is sufficient to exert this effect.

**RASSF1A-HIF-1α axis in primary NSCLC cells.** In contrast to several reports on decreased RASSF1A expression in a variety of tumors and tumor cell lines, a few reports have indicated a lack of such decrease or even an increase in RASSF1A expression in subsets of tumors of various origin[6,7,25]. We screened 56 non-small cell lung cancer (NSCLC) tissues with matched non-tumor tissues from the same lungs as controls for RASSF1A protein expression. Interestingly, in 39% of the tumor samples, we detected an increase in RASSF1A as compared to the non-tumor parts (Fig. 8a, Supplementary Fig. 11a). Further, the tumors positive for RASSF1A expression showed regions of co-immunolocalization between RASSF1A and HIF-1α as observed by immunohistochemistry (Fig. 8b) and higher fold increase in expression of hypoxic biomarkers, namely, LDHA and carbonic anhydrase 9 (CA9) (Fig. 8c). In order to establish a correlation if any between RASSF1A expression and clinical characteristics of cancer patients, we compared lung cancer pathological stage with RASSF1A expression. We observed a significant increase in expression of RASSF1A in stage III compared to stage I lung cancer patients (Fig. 8d). Next, we analyzed the RASSF1A expression in primary cancer cells isolated form the RASSF1A positive lung cancer tissue and found a strong basal expression of RASSF1A mRNA in these cancer cells as compared to A549 cell line (Supplementary Fig. 11b). To determine whether the RASSF1A-HIF-1α signaling axis is operative in these cells, we carried out HIF-1α and RASSF1A knockdown, followed by 24 h hypoxic stimulation in these primary lung cancer cells. Hypoxic exposure led to an increase in RASSF1A mRNA and protein expression in all cancer cells examined. Reduction of HIF-1α levels mediated by siRNA reversed the hypoxic increase of RASSF1A levels (Fig. 8e–i Supplementary Fig. 11c). Strikingly, knockdown of RASSF1A reduced HIF-1α levels, metabolic gene expression (LDHA, HK2: Fig. 8e–g, i–k), lactate production (Fig. 8h–l) as well as proliferation (Supplementary Fig. 11d). Together, these results suggest that RASSF1A expression is increased in a subset of non-small cell lung cancers and the primary cancer cells isolated from these tumors. Importantly, the interplay of RASSF1A-HIF-1α signaling regulates the glycolytic phenotype in these primary lung cancer cells, further supporting a broad role of RASSF1A as a hypoxia regulated protein and a crucial regulator of HIF-1α signaling.

## Discussion

The role of RASSF1A as a tumor suppressor gene, and its silencing due to methylation or mutations in different cancer cell lines has been previously established[2,25,26]. Although majorly studied in the field of malignancies, studies on its potential role in primary cells under different physiological cues such as hypoxia are unexplored. Here, we identify RASSF1A as a hypoxia regulated protein that directly promotes HIF-1α protein stabilization and transcriptional activity in various human primary cells (smooth muscle, fibroblast and cancer cells), thereby controlling the hypoxia-induced metabolic shift known as Warburg effect and cell proliferation.

Taking a step away from previous studies on RASSF1A, usually carried out in cell lines, we analyzed RASSF1A expression in different lung primary cells under hypoxia, where we noted basal expression of RASSF1A, not RASSF1C to be significantly boosted by hypoxic exposure. Further, this upregulation followed a biphasic pattern with increased protein stabilization of RASSF1A occurring as an immediate response to hypoxia, followed by prolonged transcriptional upregulation.

It is well documented that hypoxia, both acute and chronically sustained, induces a number of molecular changes with rapid and profound consequences on cell physiology[27]. Under acute hypoxic conditions (from seconds to minutes), rapid but transient changes in homeostatic mechanisms take place, primarily mediated by changes in cellular redox state and protein modifications and degradation. Particularly for the hypoxia-responsive pulmonary vascular cells, several groups have reported that low $O_2$ levels can trigger increased ROS levels with complex III of the mitochondrial electron transport chain or NOX isoforms acting as the major ROS sources[28]. Increased ROS can trigger activation of a large number of protein kinases including PKC by promoting $Ca^{2+}$ influx through voltage-dependent $Ca^{2+}$ channels[17]. By employing pan- and specific ROS inhibitors and siRNA-mediated loss of function studies, we found that the increased RASSF1A protein stability, noted within a few minutes of hypoxia exposure, is largely mediated by NOX1 derived ROS. Further, ROS activated PKCα, but not PKCβ, led to phosphorylation of RASSF1A at Ser$^{203}$ under hypoxic conditions, resulting in increased RASSF1A stabilization. The PKC-mediated Ser$^{203}$ phosphorylation data are in line with previous in vitro kinase studies, which also showed that that Ser$^{197}$ and Ser$^{203}$ of RASSF1A can be phosphorylated in response to PKCα activation in various cell lines (murine fibroblast cell line (NIH3T3), monkey fibroblast cell line (Cos-7), human embryonic kidney cells (293T))[18]. Our findings, do not exclude the possibility of parallel PKC-independent pathways. Such pathways may include phosphorylation of RASSF1A by other kinases (ATM kinase, Aurora A, Aurora B, MST1), which might further impact the RASSF1A function[29,30].

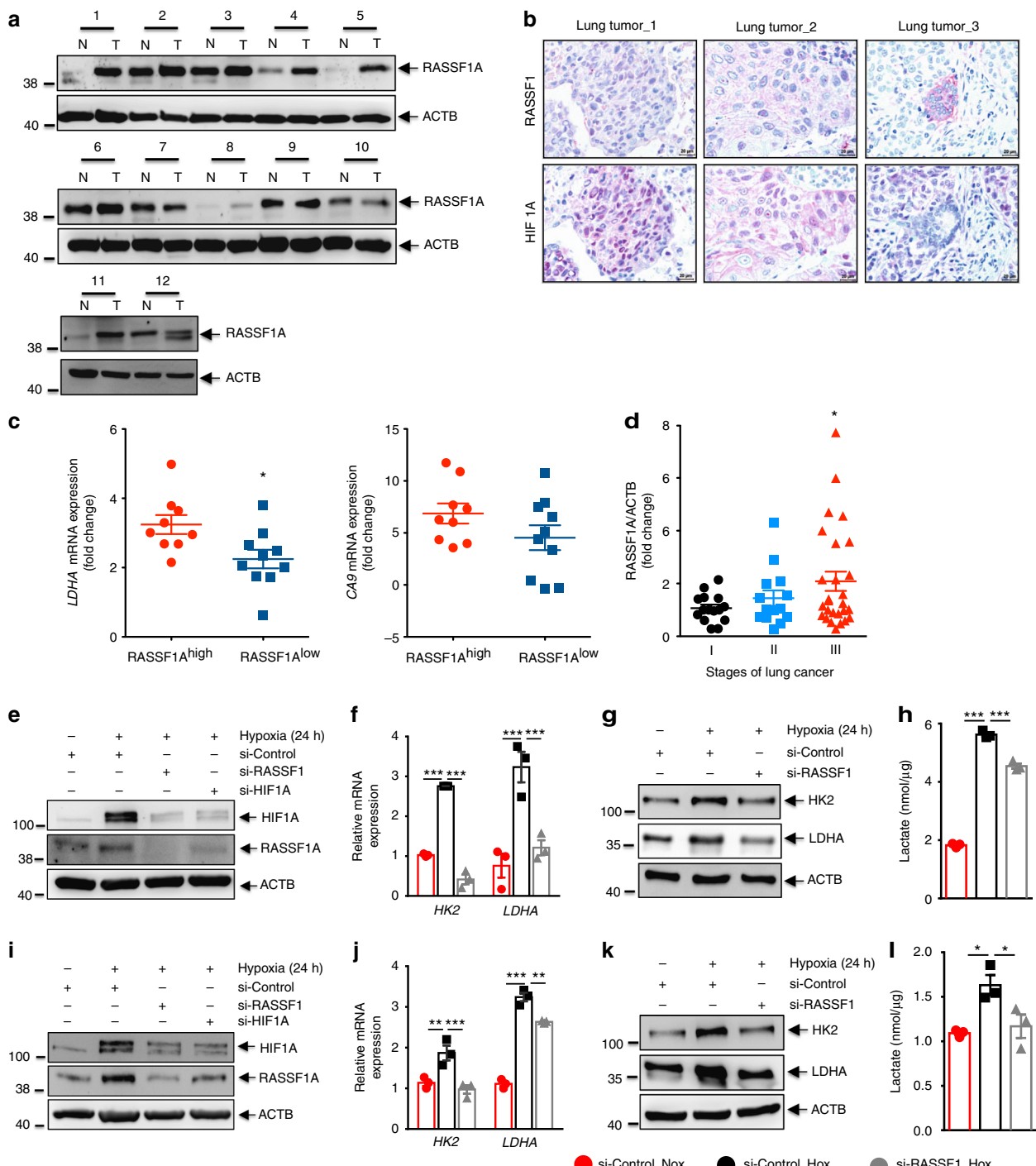

**Fig. 8** RASSF1A-HIF-1α axis in primary lung tumor cells. **a** Representative western blots for RASSF1A protein expression as analyzed in proteins isolated from tumor (T) and non-tumor (N) areas of human non-small cell lung cancer lungs. **b** Paraffin lung tissue sections from lung tumor patients were subjected to immunohistochemical staining of RASSF1 and HIF1α (HIF1A). Scale bar: 20 μm. **c** RNA was isolated from tumor and matched non-tumor samples, followed by real time PCRs for indicated genes. The Ct values of tumor samples were divided by the Ct values of respective non-tumor samples to obtain the fold change of RASSF1A expression. $n = 9$ RASSF1A$^{high}$ and $n = 10$ RASSF1A$^{low}$ tumors. **d** Fold change in RASSF1A expression in various lung tumor tissues plotted vs the pathological stages (I, II, III) of the respective tumors. $n = 15$ stage I, $n = 14$ stage II, and $n = 27$ stage III human lung tumor tissues. *$P < 0.05$ compared to RASSF1A$^{low}$ (**c**) or stage I (**d**), unpaired Student's $t$-test. **e**–**l** Primary cancer cells isolated from two patients with RASSF1A-positive lung tumor were transfected with RASSF1 siRNA (si-RASSF1), HIF1α siRNA (si-HIF1A), and control siRNA (si-Control). 24 h after transfection, cells were exposed to normoxia or hypoxia for 24 h. **e**, **i** Lysates obtained were subjected to western blotting for above-mentioned proteins. si-RASSF1 or si-control transfected lysates were further subjected to **f**, **j** real time PCRs for LDHA, HK2, **g**, **k** western blotting for LDHA, HK2 and **h**, **l** lactate assay. **f**, **h**, **j**, **l** *$P < 0.05$, **$P < 0.01$, ***$P < 0.001$ compared to si-control (hypoxia), one-way ANOVA followed by SNK multiple comparison test. Data represent mean ± s.e.m. For primary tumor cells, $n = 2$–3 independent experiments from 2 biological replicates (represented as separate)

Increased mRNA expression of RASSF1A upon prolonged hypoxia and the established role of HIF-1α as the major hypoxic transcription factor[31] led us to investigate the possible regulation of RASSF1A by HIF-1α. Interestingly, we found two HIF binding sites in the RASSF1A promoter and have shown a direct HIF binding on both of these sites, establishing RASSF1A as a HIF-1α target gene.

Overall, we identified a HIF-1-RASSF1A feed-forward mechanism, where apart from being a HIF-1α target gene, RASSF1A stabilizes HIF-1α and positively regulates its transcriptional activity in response to prolonged hypoxia. Our gain and loss of function studies of RASSF1A under hypoxic conditions indicated its role as a positive regulator of both HIF-1α protein stability and HIF-1 transcriptional activity. Further, RASSF1A overexpression augmented HIF-1α promoter occupancy and transactivation of PDK1, HK2 and LDHA in both smooth muscle cells and cancer cells. Further, Co-IP and PLA studies performed in smooth muscle cells showed a strong interaction between RASSF1A and HIF-1α. Several proteins have been previously described, which interact with and regulate the stability of HIF-1α protein via preventing its hydroxylation and proteasomal degradation. Histone deacetylase (HDAC) 7 interacts with HIF-1α and p300 to increase HIF-1 transcriptional activity[32]. STAT3 enhances HIF-1α protein stability through inhibition of pVHL binding to HIF-1α and preventing its ubiquitination[33]. It is conceivable that RASSF1A functions in a similar fashion to regulate HIF-1α stability. In line with this assumption, we found that (a) RASSF1A overexpression prevented prolyl hydroxylation of HIF-1α, (b) this was accompanied by a reduction of HIF-1α ubiquitination, and (c) binding of RASSF1A to HIF-1α negatively affected the interaction of HIF-1α with PHD2. Moreover, preliminary data suggest that RASSF1A also interacts with PHD2. All these lines of evidence indicate that RASSF1A serves as a scaffold to organize a multi-subunit complex that decreases the hydroxylation, ubiquitination and degradation of HIF-1α, thereby enhancing its transcriptional activity. Recently, Papaspyropoulos et. al. described the function of RASSF1A in stem cells differentiation which included RNA-seq data identifying hypoxia signaling downstream of RASSF1 knockdown[34], hinting in direction of our findings.

Hypoxia has been associated with major alterations in cellular phenotypes including changes in proliferation, survival and metabolic shift towards glycolysis[35–37]. In fact, effective cellular oxygen delivery is a major factor in human physiology and alterations of hypoxia-related pathways have been implicated as key drivers of multiple pathological states including cancer, cardiovascular disease, stroke, and pulmonary hypertension[38–40]. Our study identifies a crucial role of a hitherto unknown HIF-1-RASSF1A feed-forward loop in driving the hypoxia-induced cellular phenotype. Importantly, the relevance of the RASSF1A-HIF-1α axis in the setting of human disease is demonstrated by the elevated expression of RASSF1A, HIF-1α and HIF-1-target genes in pulmonary vessels of IPAH and COPD-PH patients. We also showed that RASSF1A expression is a prerequisite for the development of hypoxia-induced PH in mice. Both $Rassf1a^{-/-}$ and $Rasssf1a^{+/-}$ knockout mice were resistant to the development of PH and were unable to generate vascular remodeling and RV hypertrophic responses to hypoxia, indicating that RASSF1A mediated proliferative and metabolic alterations are required for this disease phenotype.

Strikingly, we found a subset of non-small cell lung (NSCLC) tumors (39%, RASSF1A^high) showing an upregulation of RASSF1A expression, along with co-localization with HIF-1α and increased hypoxic signatures (LDHA, CA) expression. On comparing the lung cancer pathological stage with RASSF1A expression, we observed a significant increase in the expression of RASSF1A in stage III compared to stage I lung cancer patients. In nearly 50% of the stage III lung cancer patients a marked increase in RASSF1A expression was noted, which may suggest that RASSF1A^high lung cancers are particularly aggressive and/or have a worse prognosis via regulation of HIF-1α and metabolic switch. Further, basal expression of RASSF1A transcript was observed in primary cancer cells isolated from these tumors, which was strongly boosted by exposure to hypoxia. Gene silencing studies in these cells with siRASSF1 and siHIF-1α under hypoxic conditions resembled the findings in hypoxic vascular cells, verifying that the RASSF1A-HIF-1α axis regulates the metabolic shift and lactate production also under conditions of malignancy. Interestingly, our findings correspond to a previous report describing equal frequency of increase (16/38, 42%) and decrease of RASSF1A mRNA expression in lung epithelial cancer cells. The authors also showed that in renal cell carcinomas the frequency of RASSF1A mRNA expression increase was even higher (24/38, 63%)[7]. Another study also carried out in renal cell carcinoma demonstrated that tumor cells expressing RASSF1A showed increased tumor progression[6]. Hence, our study taken together with these reports suggests that non-small cell lung cancer presents as 2 phenotypes, RASSF1A^low where RASSF1A is epigenetically or genetically silenced[26,41], and RASSF1A^high where RASSF1A expression is maintained and even increased in response to hypoxic microenvironment. Several previous studies suggested that RASSF1A has a tumor suppressor function during early tumor development, and that its silencing due to hypermethylation leads to activation of proliferative and migratory processes via inactivation of Hippo signaling[42] and cyclins[43] and RAS induced apoptosis[44]. In line with these observations, a prognostic value of RASSF1A downregulation in lung cancer was noted. However, though, strongly prevalent, hypermethylation is not observed in all tumors. Some tumors do not exhibit epigenetic or genetic silencing of RASSF1A, but it may even be increased (RASSF1A^high), and in this subset of tumors RASSF1A will promote the Warburg effect via HIF1α. We here show that this is linked with a hypoxic microenvironment, NOX1 expression and activity and HIF1α expression. Interestingly, both NOX1 and HIF1α expression have displayed positive correlation with clinical stages in NSCLC[45,46].

Collectively, we report a hitherto unrecognized crucial role of RASSF1A in regulating HIF-1α to promote hypoxia-driven gene regulation, metabolic switch and hyperproliferation in pulmonary hypertension as a non-malignant hypoxia-induced prototype disease and lung cancer. The underlying molecular mechanisms unveiled here (Fig. 9) provide future targets for therapeutic intervention, to be exploited for improved therapy of these diseases.

## Methods

**Human lung samples**. Human explanted lung tissues from subjects with IPAH or control donors were obtained during lung transplantation. Samples of donor lung tissue were taken from the lung that was not transplanted. The study protocol for tissue donation was approved by the ethics committee (Ethik Kommission am Fachbereich Humanmedizin der Justus Liebig Universität Giessen) of the University Hospital Giessen (Giessen, Germany) in accordance with national law and with Good Clinical Practice/International Conference on Harmonisation guidelines. Written informed consent was obtained from each individual patient or the patient's next of kin (AZ 58/15).

Biomaterials and data from NSCLC patients were provided by the Lung Biobank Heidelberg, a member of the accredited Tissue Bank of the National Center for Tumor Diseases (NCT) Heidelberg, the BioMaterial Bank Heidelberg and the Biobank platform of the German Center for Lung Research (DZL). All subjects gave their informed consent for inclusion before they participated in the study. The study was conducted in accordance with the Declaration of Helsinki. The use of biomaterial and data for this study was approved by the local ethics committee of the Medical Faculty Heidelberg (S-270/2001).

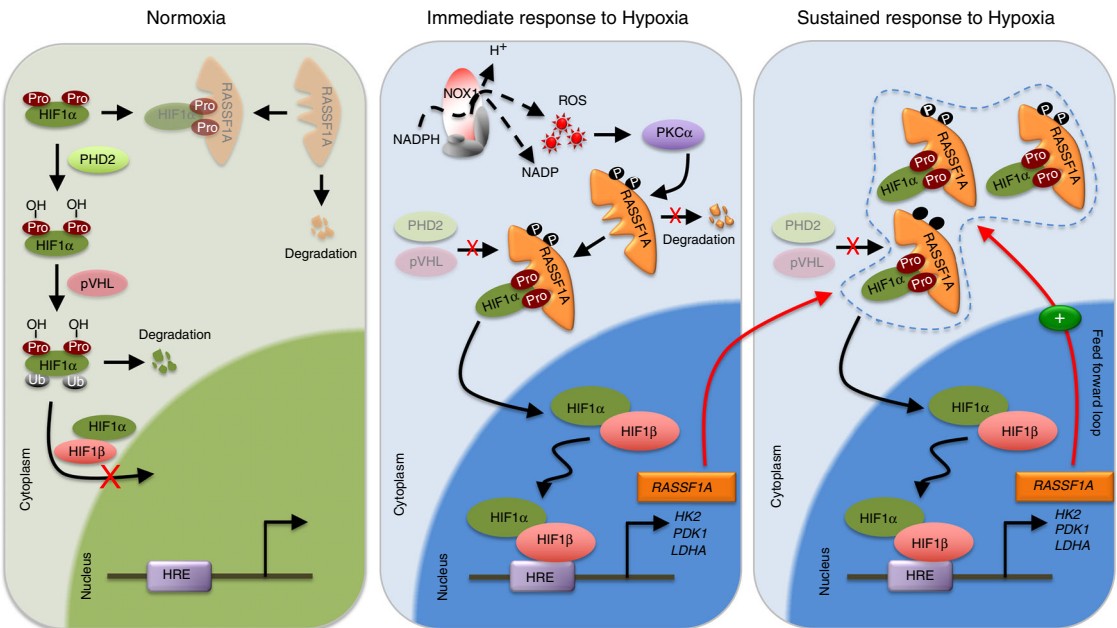

**Fig. 9** Schematic depicture of RASFF1A mediated HIF regulation. Under normoxia, HIF1α is hydroxlyated at proline residues by PHDs, ubiquitinated, followed by proteasomal degradation. Under hypoxia, RASSF1A is phosphorylated by ROS activated PKCα, leading to increased stability. Increased RASSF1A binds to HIF1α, preventing its binding to PHD2 and prolyl hydroxylation, increased nuclear translocation and subsequent transcriptional activity. This in turn leads to increased expression of glycolytic genes (PDK1, HK2, LDHA) and RASSF1A itself, giving rise to a feed forward loop and increased proliferation and glycolysis, manifesting in pulmonary hypertension and lung cancer pathogenesis. RASSF1A: Ras association domain family 1A, pVHL: von Hippel-Lindau tumor suppressor protein, PHD2/3: prolyl hydroxylase 2/3, HIF1α: hypoxia-inducible factor 1 alpha, HIF1β: hypoxia-inducible factor 1 beta, HRE: hypoxia-response elements, Ub: ubiquitin, OH: hydroxylation, Pro: proline, NADPH: nicotinamide adenine dinucleotide phosphate hydrogen, ROS: reactive oxygen species, NOX1: NADPH oxidase 1, NADP: nicotinamide adenine dinucleotide phosphate, PKCα: protein kinase C alpha, P: phosphorylation, HK2: hexokinase 2, PDK1: pyruvate dehydrogenase kinase, isozyme 1, LDHA: lactate dehydrogenase A

**Cell culture**. Human PASMCs was obtained from Lonza and grown in SmGM-2 Bulletkit medium (Lonza). Human PAAFs were obtained from ScienCell and grown on poly-lysine (Sciencell) coated dishes in fibroblast medium (ScienCell). Cells were maintained at 37 °C in a 5% CO2 incubator. Experiments were performed with cells from passages 6 to 7.

Explant-derived PASMCs were obtained from human pulmonary arteries (<2 mm in diameter) obtained from patients with IPAH. Segments of pulmonary artery were cut to expose the luminal surface. The endothelium was removed by gentle scraping with a scalpel blade, and the media was peeled away from the underlying adventitial layer. The medial explants were cut into ~1- to 2-mm² sections, transferred to T75 flasks with Promocell smooth Muscle Cell Growth Medium 2 (Promocell) and allowed to adhere for 2 h at 37 °C and 5% CO₂. Once the cells had adhered, the explants were incubated in DMEM supplemented with 20% FBS until cells had formed confluent monolayers. PASMCs were trypsinized, and subsequent passages were cultured in smooth muscle cell growth medium 2 and maintained at 37 °C in 5% CO₂.

Primary tumor cells were obtained from University of Giessen and Marburg Lung Center (UGMLC) biobank and Lung Biobank Heidelberg, a member of the accredited Tissue Bank of the National Center for Tumor Diseases (NCT) Heidelberg, the BioMaterial Bank Heidelberg, and the Biobank platform of the German Center for Lung Research (DZL). Tumor cells were grown on collagen IV (Sigma) coated dishes in DMEM/Ham's F12 medium with different supplements (Sodium selenite, Ethanolamine, Phosphorylethanolamine, Sodium pyruvate, Adenine: all from Sigma, Airway epithelial growth medium supplement pack: Promocell, ROCK inhibitor: Selleckchem). Experiments were performed with cells from passages 5 to 8.

Various cell lines (HEK 293 and HeLA) were obtained from ATCC and cultured in DMEM medium containing 10% fetal calf serum and 1% antibiotics.

**Hypoxia exposure of human PASMCs and PAAFs**. Hypoxia experiments were performed in a chamber equilibrated with a water-saturated gas mixture of 1% O₂, 5% CO₂, and 94% N₂ at 37 °C. Human PASMCs and human PAAFs were incubated in hypoxia or normoxia chambers for indicated time points in hypoxic medium (basal medium containing 1% FCS for human PASMCs and human PAAFs, serum free medium for tumor cells).

**Transfection with siRNA**. Human PASMCs and tumor cells were transfected with different siRNAs using Lipofectamine 3000 Transfection Reagent (Invitrogen) in

optiMEM serum free medium. As a control, commercially available non-targeting siRNA (si-Control) was used. All siRNAs were obtained from Qiagen (Supplementary Table 2). Six hours after transfection, cells were cultured in serum-containing medium for a resting period of 24 h, followed by hypoxia exposure for different time points. Transfection of human PAAFs was carried out with siRNA targeting RASSF1 by electroporation using A-024 program of primary fibroblasts nucleofection kit (Amaxa, Lonza) according to manufacturer's instructions.

**Transfection with plasmids**. pCMV tag1-RASSF1A, pCMV tag1-RASSF1A S203D, pCMV tag1-RASSF1A S203A, pCMV tag1-RASSF1C and empty vector were provided by Prof. R. Dammann. pGL3-HRE plasmid was provided by Dr. Savai. p2xFLAG-hYAP1 (#17791), p2xFLAG-YAP1-S127A (#17790) and 8xGTIIC-luciferase (#34615) were obtained from Addgene. 180 bp and 2100 bp region from RASSF1A promoter were PCR amplified from human lung genomic DNA and cloned into pGL3 vector upstream of firefly luciferase gene. Human PASMCs and human PAAFs were transfected with plasmids using A-033 program of primary smooth muscle cells nucleofection kit and A-024 program of primary fibroblasts nucleofection kit respectively (Amaxa, Lonza). Six hours after transfection, medium was changed to serum-containing medium for a resting period of 24 h, followed by hypoxia exposure for different time points. HEK 293 cells were transfected with plasmids using turbofect (Thermo Scientific) directly in serum-containing medium. 24 h later, cells were exposed to hypoxia for different time points.

**Treatment with compounds**. Human PASMCs were incubated with hypoxia medium (1% FCS) in the absence or presence of N-acetyl cysteine (0.5 or 1 mM: Sigma-Aldrich), DPI (5 or 10 μM), rotenone (5 or 10 μM: Sigma-Aldrich), TTFA (5 or 10 μM: Sigma-Aldrich), antimycin A (5 or 10 μM: Sigma-Aldrich), sodium azide (0.5 or 1 mM), 3-Nitro propionic acid (3 or 5 mM: Sigma-Aldrich)Ku55933 (5 or 10 μM: Sigma-Aldrich), GKT137831 (5 or 10 μM) Gö6976 (0.5 or 1 μM: Roche) for 1 h followed by 15 min of hypoxia exposure. Relative change in RASSF1A protein expression was assessed by western blotting. The concentration of the compounds was chosen based on previous studies.

**Immunocytochemistry**. Human PASMCs grown on chamber slides were treated and fixed with acetone-methanol (1:1), washed 3 times for 5 min with PBS and blocked for 1 h with blocking buffer (5% BSA, 0.5% goat serum, 0.2% Triton-X in PBS), and incubated overnight with a RASSF1 primary antibody (1:100; abcam)

overnight. This was followed by 1 h incubation with secondary antibody Alexa Fluor®-488 (1:1000, Life Technologies, A11008). After incubation, slides were counterstained with ToPro3 (for nuclear staining) and mounted with fluorescent mounting medium (Dako). Fluorescent images were taken with LSM 710 confocal microscope.

**Proximity ligation assay.** Human PASMCs were grown on chamber slides and exposed to hypoxia for different time points. After the exposure, cells were fixed with acetone: methanol (1:1), followed by Duolink® (Sigma Aldrich) proximity ligation assay according to manufacturer's protocol employing HIF1A and RASSF1A antibody. Fluorescent images were taken with LSM 710 confocal microscope.

**Assessment of proliferation of primary cells.** The influence of RASSF1A knockdown and overexpression on proliferation was assessed with BrdU incorporation assay (Roche Diagnostics) according to manufacturer's instructions. Absorbance was measured at 370 nm with reference at 492 nm in a plate reader (TECAN, Germany). Proliferation of cells was plotted as a percentage of absorbance compared to control cells absorbance.

**Coimmunoprecipitation (Co-IP).** HEK293 cells were grown to 80% confluence in 100 mm dishes and then, transfected with various plasmids (RASSF1A-FLAG, RASSF1A-FLAG (S203A), RASSF1A-FLAG (S203D) or HIF1A) using Turbofect (Thermo Scientific). 6 h after transfection, cells were subjected to 24 h hypoxia or normoxia, followed by cell lysate preparation in Co-IP lysis buffer (Thermo Scientific). Addtionally, for imunoprecipitations carried out to check post-translational modifications (prolyl-hydrxylation, lys48 ubiquitination) of HIF1α, cells were pretreated with MG132 (Santa Cruz) for 30 min before hypoxia exposure, followed by 6 h hypoxia exposure and lysis in Co-IP lysis buffer. Lysates were centrifuged and supernatants were further used for CoIP. Equal amounts of proteins were incubated overnight on rotation with anti FLAG (Sigma), anti-HIF1A (abcam), anti-PHD2 (Novus biological) or IgG (Santa Cruz), followed by 3 h incubation with Protein G sepharose beads (GE healthcare). After incubation, beads were repeatedly washed with PBS + 0.1% tween20, followed by addition of SDS sample buffer and analysis by western blotting.

**Chromatin-immunoprecipitation (ChIP).** Cells (human PASMCs or HEK293) were exposed to 20% or 1% O2 for 24 h, cross-linked with 1% formaldehyde for 10 min at room temperature, and quenched in 0.125 M glycine. DNA was immunoprecipitated from the sonicated cell lysates using anti-HIF1A antibody (Abcam) and immunoprecipitates were recovered by addition of Salmon Sperm DNA/Protein A Agarose-50% Slurry. After washing, elution, and reverse cross-linking, DNA was purified by PCR purification kit (Qiagen). Purified DNA was quantified by SYBR Green real-time PCR (Bio-Rad) using specific primers (Supplementary Table 1). Data is expressed as percentage of input, calculated from the formula: **% of Input is equal to $2^{(-dCt)}$, dCt is Ct ChIP – (Ct Input – log$_2$ dilution factor)**. The average from IgG control was set to 1.

**Lactate assay.** HEK 293 cells were grown to 80% confluence and then transfected with RASSF1A or empty vector. 24 h after transfection, cells were exposed to 24 h hypoxia or normoxia, followed by measurement of lactate production using lactate assay kit (Abcam) according to manufacturer's protocol. Absorbance was measured at 450 nm in a plate reader (TECAN). Lactate concentrations were determined on basis of the lactate standards and normalized to total protein content of each sample. Tumor cells were transfected with siRNA against RASSF1 and Control siRNA as described before and 24 h after hypoxia exposure, lactate production was measured.

**Reporter gene assays.** HEK293/HeLa cells were grown to 80% confluence in 48 well plates and then co-transfected with HRE luciferase construct or an empty pGL3 vector (Promega), various expression vectors mentioned in the text or RASSF1 siRNA and the internal control pRL-CMV vector (Promega). 6 h after transfection, cells were exposed to hypoxia or normoxia for further 24 h before the preparation of cell lysates. Both firefly and *Renilla* luciferase activities were quantified using the dual-luciferase reporter assay system (Promega) according to the manufacturer's instructions and employing a spectrofluorometer (BioTek Instruments). The ratio of luciferase signal to *Renilla* signal for each well was calculated. The average from control samples was set to 1.

**Promoter analysis.** Sense and antisense strands of the human RASSF1 promoter were screened upstream and downstream of the coding sequence of the RASSF1 gene (NM_007182.4) for potential HIF-1 binding sites (HBS:G/ACGTG).

**Animal experiments.** The experiments were performed in accordance with the US National Institutes of Health Guidelines on the Use of Laboratory Animals. Both the University Animal Care Committee and the federal authorities for animal

research of the Regierungspräsidium Giessen and Darmstadt (Hessen, Germany) approved the study protocol (B2/325). Mice (male, 12–14 weeks old) were used for experiment.

**Genotyping.** Mice carrying targeted alleles were genotyped by PCR. For genotyping mouse RASSF1A lines, primers used were: Primer1 5′-TTGTGCCGTGC CCCGCCCA-3′; Primer2 5′-TGACCAGCCCTCCACTGCCGC-3′ and Primer3 5′-GGGCCAGCTCATTCCTCCCAC-3′. DNA was extracted from mouse tails, and 4 µl was used in the subsequent PCR reaction. Cycling conditions were 5 min at 95 °C; 35 cycles of 30 s at 95 °C, 20 s at 64 °C and 60 s at 72 °C; followed by 7 min at 72 °C. The WT and knock out alleles resulted in 520-bp and 380-bp bands, respectively.

**Hypoxia exposure of mice.** Hypoxic pulmonary vascular remodeling was induced by exposure of mice to chronic hypoxia (10% O2) in a ventilated chamber, as previously described[47]. Briefly, animals were age-matched and randomly distributed to groups exposed to normoxia or chronic hypoxia (28 days).

**MRI, hemodynamic, and RV hypertrophy measurements.** Cardiac MRI measurements were performed on a 7.0T Bruker Pharmascan, equipped with a 380 mT/m gradient system, using a custom-built circularly polarized birdcage resonator and the IntraGateTM self-gating tool (Bruker, Ettlingen, Germany). The parameters for identification of the ECG were adapted for one heart slice and transferred afterwards to the navigator signals of the remaining slices. Thus the in-phase reconstruction of all pictures is guaranteed[48]. The 14 weeks old mice were measured pre and 4 weeks post hypoxia under volatile isoflurane (1.5–2.0%) anesthesia. The measurement is based on the gradient echo method (repetition time = 6.2 ms; echo time = 1.6 ms; field of view = $2.20 \times 2.20$ cm; slice thickness = 1.0 mm; matrix = $128 \times 128$; repetitions = 100). The imaging plane was localized using scout images showing the 2- and 4-chamber view of the heart, followed by acquisition in short axis view, orthogonal on the septum in both scouts. Multiple contiguous short-axis slices consisting of 7–10 slices were acquired for complete coverage of the left and right ventricle. MRI data were analyzed using Qmass digital imaging software (Medis, Leiden, Netherlands).

Hemodynamics in KO and WT mice were measured as described previously[49]. Briefly, anesthesia was given to mice as described above. After intubation, the mouse was placed in a supine position on a homeothermic plate (AD Instruments, Spechbach, Germany) and connected to a small-animal ventilator (MiniVent type 845, Hugo Sachs Elektronik, March-Hugstetten, Germany). Body temperature was controlled by a rectal probe connected to a control unit (AD Instruments) and was kept at 37 °C during catheterization. The right external jugular vein was catheterized with a high-fidelity 1.4F micromanometer/Mikro-Tip Pressure catheter (Millar Instruments, Houston, TX) and advanced into the RV to assess RVSP. Subsequently, the 1.4F micromanometer catheter was inserted into the aorta and the LV through the left carotid artery for measurement of SAP. Data were collected and analyzed using the PowerLab data acquisition system (MPVS-Ultra Single Segment Foundation System, AD Instruments) and LabChart 7 for Windows software.

After exsanguination, the left lung was fixed for histology in 10% neutral buffered formalin, and the right lung was snap frozen in liquid nitrogen. For right-heart hypertrophy, the RV was separated from the LV plus septum, and the RV/tibia length ratio was determined from the tissue.

**Immunofluorescence staining and assessment of proliferation.** Paraffin-embedded lung tissue sections (3-µm thick) were deparaffinized in xylene and rehydrated in a graded ethanol series to PBS (pH 7.2). Antigen retrieval was performed by pressure cooking in citrate buffer (pH 6.0) for 15 min. Double immunofluorescence staining was performed with primary antibodies to RASSF1 (1:200, ab52857, Abcam Ltd., Cambridge, UK), HIF1A (1:200), α-actin (1:400, A2547, Sigma, Saint Louis, MO), vWF (1:200, IS527, Dako) and collagen I (1:200, c2456, Sigma). After overnight incubation, slides were washed and incubated with the respective secondary antibodies, Alexa 488- and Alexa 555-conjugated goat anti-rabbit IgG (1:1000, Molecular Probes) for 1 h. All sections were counterstained with nuclear DAPI (1:1000) and mounted with fluorescent mounting medium (Dako).

Tissue sections were also stained for PCNA (1:100, sc-7907, Santa Cruz Biotechnology) to detect proliferation. PCNA positive pulmonary vascular cells were counted throughout the entire section and expressed as fold change in percentage of the control lung by calculating the number of PCNA-positive cells per pulmonary vessel.

**Medial wall thickness measurement.** For morphometric analysis, Hematoxylin and eosin staining was performed according to common histopathological procedures. Analysis was done in a blinded fashion. To assess the type of remodeling of muscular pulmonary arteries, microscopic images were analyzed using a computerized morphometric system (QWin; Leica). Media thickness was defined as the distance between the lamina elastica interna and the lamina elastica externa.

Depending on the external diameter of the pulmonary arteries, they were categorized as follows. Category I included arteries with an external diameter of 25–70 μm and category II included arteries with an external diameter of 70–150 μm and category III included arteries with external diameter greater than 150 μm.

**Laser-assisted microdissection of pulmonary vessels**. Laser-assisted microdissection of eight donor lungs and eight lungs from patients with IPAH or COPH-PH was performed as described. 100–150 arteries per patient were collected; cryosections from lung tissues were mounted on glass slides. After brief staining with hemalaun, intrapulmonary vessels were microdissected from the sections with the use of the Laser Microbeam System (P.A.L.M, Bernried, Germany) as described[50]. Total cellular RNA from vessels that adhered to the dissecting needles was isolated with the micro RNeasy kit (Qiagen).

**RNA isolation, cDNA synthesis, and qPCR**. Total mRNA was extracted from frozen human pulmonary arteries and various cells with the RNeasy Mini kit (Qiagen) or Trizol (Life technologies) respectively. Equal amounts of isolated RNA were subsequently transcribed into cDNA using High-capacity cDNA reverse transcription kit (Applied Biosystems) according to the manufacturer's instructions. qPCR was then performed with the iQ SYBR Green Supermix (Bio-Rad) kit. Intron-spanning human-specific, mouse-specific and rat-specific primers for the mentioned genes, were designed using sequence information from the NCBI database and were purchased from Metabion (Martinsried, Germany). Expression was analyzed with the ΔCt method. The Ct values of the target genes were normalized to that of the housekeeping gene (endogenous control) encoding beta-2-microglobulin (*B2M*) or hypoxanthine-guanine phosphoribosyltransferase (*HPRT*) using the equation $\Delta Ct = Ct_{reference} - Ct_{target}$ and expressed as ΔCt. The primers used in this study are shown in Supplementary Table 1. Relative mRNA expression is also shown, with the average from control samples set as 1.

**Western blotting and quantification**. Lung tissue samples, pulmonary artery samples and cells were homogenized in RIPA lysis buffer (Thermo scientific), quantified and lysates were separated on 10% polyacrylamide gels and transferred to nitrocellulose membranes. After blocking, the membranes were probed with one of the following antibodies: anti-HIF1A (1:1000, BD Biosciences), anti-HIF2A (1:1000, Novus Biologicals), anti-ACTB as a loading control (1:5000, NB600–502, Novus Biologicals) and anti-RASSF1A (1:500, ab23950), anti-HK2 (1:2000, ab9464), anti-PDK1 (1:500, ab9461) and anti-LDHA (1:1000) all from Abcam Ltd. This was followed by 1 h incubation with secondary antibodies conjugated with horseradish peroxidase (HRP). Bound antibodies were detected by chemiluminescence with the ECL detection system (Thermo Scientific,) using Image reader (GE Healthcare) and densitometric analysis of the blots was obtained using multi gauge software (Fujifilm, Tokyo, Japan). In some experiments, Image reader (Fujifilm) was used for visualizing and quantifying western blot bands. Expression was quantified using band intensity values (in arbitrary units), which were normalized to ACTB. Uncropped images of all western blots are provided in the Supplementary Figs. 12–27. Detailed information of antibodies is provided in Supplementary Table 3.

**Statistical analysis**. All data are expressed as mean ± standard error of the mean (s.e.m.). Statistical comparisons of samples were performed by Student's *t* test for comparing two groups or one-way ANOVA followed by the Student–Newman–Keuls (SNK) post-hoc test for multiple comparisons. Grouped comparisons were carried out by two-way ANOVA. Difference with $P < 0.05$ between the groups was considered significant. All statistical analyses were performed using Prism 5.0 and Prism 6.0 (GraphPad Software).

**Reporting summary**. Further information on research design is available in the Nature Research Reporting Summary linked to this article.

## Data availability
The data supporting the findings of this study are available from the corresponding author upon request.

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

## Acknowledgements

The authors would like to thank Prof. Reinhard Dammann, Institute for Genetics, Justus-Liebig University Giessen for excellent suggestions with regard to RASSF1A and providing all the RASSF1A and RASSF1C constructs; Prof. Joachim Fandrey, Institute of Physiology, University of Duisburg-Essen for support with cell lines for RASSF1screening. The authors thank Uta Eule, Vanessa Golchert, Ewa Bieniek, and Natascha Wilker for the valuable technical assistance. The Max Planck Society, the Scientific and Economic Excellence in Hesse (LOEWE) Program, DFG, SFB 1213 (Project A01, A05, A06, A07), and the Excellence Cluster 147 Cardio-Pulmonary System (ECCPS, EXC 147) supported this work.

## Author contributions

S.D., C.M., R.S., C.V., and M.S. acquired the data. S.D., R.S., C.S., C.V., A.W., and S.S.P. analyzed and interpreted the data. S.D., W.S., and S.S.P. conceived and designed research. G.L.S. and N.W. contributed HIF constructs and NOX siRNAs. M.M., T.M., and T.S.-N. have provided primary tumor cells and microdissected human tumor vs non-tumor samples. S.D., C.S., N.W., W.S., F.G., and S.S.P. drafted the manuscript. W.S. and S.S.P. handled the funding and supervision.

## Additional information

**Competing interests:** The authors declare no competing interests.

