## [Peer Review File · Nature Communications]

Reviewers' comments:

Reviewer #1 (Remarks to the Author):

The manuscripts by Dabral et al. reports that the tumour suppressor RASSF1A is a scaffold for Hif1 stabilisation upon hypoxia. They report found that hypoxia induces RASSF1A protein stabilization and that after prolonged hypoxia exposure, HIF-1 α activates RASSF1A transcription. They outline how RASSF1A levels support the Warburg effect through the classical metabolic/glycolytic switch and is involved in vascular disease and pulmonary hypertension (PH). The manuscript is very novel, highly relevant and comprehensive although I believe there are mainly conceptual points and important controls that are required to be addressed.

Major.

1. The authors conclude that this new role for RASSF1A in PH and vascular disorders also supports a role in a subset of NSCLC. They rightly conclude that their results suggest that RASSF1 low tumours are a distinct phenomenon. However, the manuscript goes too far in dismissing the tumour suppressor role for RASSF1A as something that a 'Epigenetic silencing (promoter methylation) and genetic changes (somatic mutations) are observed in various cancers and in particular human cancer cell lines and RASSF1A was hence suggested to function as tumor-suppressor (Richter et al., 2009)'. This is entirely misleading, there is an enormous quantity of studies not just supporting, but qualifying, the tumour suppressor role in all solid malignancies. This is intentionally misleading to support a novel mechanism and not necessary. The authors do show a very novel aspect that could be important in cancer and do not need to dismiss the importance of RASSF1A silencing. After all there is substantial evidence that this is not just an anecdotal, but that it is prognostic and predictive. Moreover, there are numerous meta-analyses confirming these effects (both epigenetic and genetic) not just in lung.

2. The authors have not described the antibodies and siRNA well enough for the reader to know if they are targeting RASSF1A or RASSF1C. Nor whether HIF1 promotes expression of RASSF1A or 1C or both. Without the experiments clarifying whether the oncogenic version is expressed by HIF1 or involved in the stabilisation, they cannot entirely conclude that this is a RASSF1A specific effect.

3. RASSF1A is a key regulator of hippo pathway signalling, yap/taz have also been implicated in HIF1 activation. This needs to be addressed to determine the full picture of the regulation.

4. Are RASSF1^{high} associated with hypoxia signatures? Are these particularly aggressive or have a worse prognosis as would be expected of hypoxic tumours?

5. From what I can determine it appears that the RASSF1 upregulation may be an important protective mechanism in coping with hypoxia, as HIF1 activation is not necessarily a dangerous event. However, in the absence of RASSF1A this may not function as well and lead to promotion.

In all I really like the study and am supportive. I would encourage the authors to fully explain their findings in context (which do nicely fit) rather than being dismissive. Statements like 'Classified as a

tumor suppressor, RASSF1A has predominantly been studied in cancer cell lines and transformed cells.' Implies everyone has been mistaken. There is far more clinical and in vivo data actually published than the work presented here. RASSF1A has been explored in HBECS and other normal cells, albeit not under hypoxia, but this displays a lack of informed opinion though not being aware of the literature.

The authors also write 'In contrast to several reports on decreased RASSF1A expression in a variety of tumors and tumor cell lines, some reports have indicated the lack of such decrease or even an increase in RASSF1A expression in subsets of tumors of various origin (Palakurthy et al., 2009; Pronina et al., 2012; Tezval et al., 2008).' Several reports? Try >1500 articles! This is not acceptable behaviour to increase the potential impact of your study. Especially we all you screened was '25 non-small cell lung tumor tissues', compared to the literal 10 of 1000s of samples. Again, this is unnecessary in an otherwise very nice study, and needs substantial amendment.

[Note from the Editor: reviewer #1 expressed in further correspondence that full western blots including molecular weight markers should be provided to confirm specific targeting of RASSF1A (point #2).]

Reviewer #2 (Remarks to the Author):

In this manuscript, Debrai and colleagues report their observations on the role of RASSF1A-HIF1A in hypoxia-induced proliferation and glycolysis in cell cultures, in human tissues and a mouse model of hypoxic pulmonary hypertension (PH), and in human tissues and cell cultures of lung cancer specimens. The overall finding that RASSF1A strongly regulates HIF1A function and expression in hypoxia is a little outside the paradigm of direct hypoxic regulation of HIF1a, but the data are generally quite strong and this fits with an evolving literature of alternative regulators of HIF1a. Overall the manuscript is very well written and flows logically, with the figures clear.

Major

1. One of the key components in the work is in regards to how the phosphorylation of RASSF1A in the setting of hypoxia regulates RASSF1A stability and function, but this is presently not very well developed. What are the effects of RASSF1A phosphorylation on its stability over time (aside from direct expression levels as shown 2g), and the physical or functional interaction between it and HIF1a?

2. Can the authors determine the subcellular location of the expression (and ideally the site of its function) of RASSF1A in the cytoplasm versus nucleus? As noted, HIF1a stabilization primarily is a cytoplasmic event, with subsequent translocation to the nucleus and transcriptional function. One of the notable findings in the immunostaining (particularly the human tissue in figure 5b) is the diseased specimens have an increase in the relative expression of RASSF1 in the cytoplasm, but the nuclear expression of RASSF1 doesn't change much.

3. In the studies of the phenotype of the PSMCs from IPAH and donors (such as brdu uptake in figure 5f), it would ideal to perform dual knockdown of RASSF1A and HIF1a (as previously done in figure 2) to determine how much of the IPAH cell phenotype induced by RASSF1A is mediated by HIF1a.

4. Does knockdown of RASSF1 in the lung cancer cells affect the cell phenotype, such as rate of proliferation or BrdU incorporation?

Minor

5. It would be ideal, but is not critical, to performed immunostaining of the hypoxic PH mouse lung tissue for RASSF1, similar to the panel in figure 5b.

Reviewer #1 (Remarks to the Author):

The manuscript by Dabral et al. reports that the tumor suppressor RASSF1A is a scaffold for Hif1 stabilization upon hypoxia. They report found that hypoxia induces RASSF1A protein stabilization and that after prolonged hypoxia exposure, HIF-1 α activates RASSF1A transcription. They outline how RASSF1A levels support the Warburg effect through the classical metabolic/glycolytic switch and is involved in vascular disease and pulmonary hypertension (PH). The manuscript is very novel, highly relevant and comprehensive although I believe there are mainly conceptual points and important controls that are required to be addressed.

Major:

1. The authors conclude that this new role for RASSF1A in PH and vascular disorders also supports a role in a subset of NSCLC. They rightly conclude that their results suggest that RASSF1A low tumors are a distinct phenomenon. However, the manuscript goes too far in dismissing the tumor suppressor role for RASSF1A as something that a 'Epigenetic silencing (promoter methylation) and genetic changes (somatic mutations) are observed in various cancers and in particular human cancer cell lines and RASSF1A was hence suggested to function as tumor-suppressor (Richter et al., 2009)'. This is entirely misleading, there is an enormous quantity of studies not just supporting, but qualifying, the tumor suppressor role in all solid malignancies. This is intentionally misleading to support a novel mechanism and not necessary. The authors do show a very novel aspect that could be important in cancer and do not need to dismiss the importance of RASSF1A silencing. After all there is substantial evidence that this is not just an anecdotal, but that it is prognostic and predictive. Moreover, there are numerous meta-analyses confirming these effects (both epigenetic and genetic) not just in lung.

R1. We thank the reviewer for the comment and would like to mention that it was by no means our intention to dismiss the established role of RASSF1A as a tumor suppressor gene. More than a thousand studies have convincingly presented promoter methylation, point mutations and decreased expression of RASSF1A in varied kinds of malignancies, strongly directing towards its prognostic usefulness. Consequently, we have replaced the sentence 'Epigenetic silencing (promoter methylation) and genetic changes (somatic mutations) are observed in various cancers and in particular human cancer cell lines and RASSF1A was hence suggested to function as tumor-suppressor (Richter et al., 2009)' with 'Epigenetic silencing (promoter methylation) and genetic changes (somatic mutations) are observed in various cancers and in particular human cancer cell lines (PMID: 11313894; PMID: 28123306; PMID: 12360410), establishing RASSF1A as a bonafide tumor-suppressor (Richter et al., 2009)'.

2. The authors have not described the antibodies and siRNA well enough for the reader to know if they are targeting RASSF1A or RASSF1C. Nor whether HIF1 promotes expression of RASSF1A or 1C or both. Without the experiments clarifying whether the oncogenic version is expressed by HIF1 or involved in the stabilization, they cannot entirely conclude that this is a RASSF1A specific effect.

R2: As requested by the reviewer, we have added the information regarding the antibodies used for RASSF1A, along with all others in the revised manuscript in supplementary table 3. Furthermore, the siRNA used in our manuscript is a commercially available siRNA (Qiagen) targeting a sequence common to both isoforms, RASSF1A and RASSF1C. Thus, the siRNA has always been described as si-RASSF1 in the manuscript.

In order to confirm whether the effect of si-RASSF1 on HIF-1 α is mediated via RASSF1A or RASSF1C or both, we have carried out a rescue experiment where PSMCs treated with RASSF1 siRNA were further subjected to RASSF1A or RASSF1C overexpression and exposed to hypoxia. Interestingly, only RASSF1A overexpression was able to rescue the effect of RASSF1 knockdown on HIF-1 α protein expression and subsequent HIF target gene mRNA expression. The result is now provided in the Supplementary Fig. 6 c-d of the revised manuscript. This result is in agreement with results previously presented in the manuscript, where we observed HIF-1 α stabilization and elevated transcriptional activity under RASSF1A overexpression (Fig 4b-d).

Supplementary Fig. 6: Human PSMCs were transfected with RASSF1 siRNA (si-RASSF1) and control siRNA (si-Control). 24 hr later, cells were further transfected with EV, RASSF1A-FLAG or RASSF1C-FLAG as mentioned in the lanes above the blots. 6 hr later, cells were exposed to Hypoxia or Normoxia for further 24 hr. **(c, left)** Cell lysates were subjected to western blotting for indicated proteins, followed by **(c, right)** densitometric quantification of HIF1A/ACTB expression ratio. ACTB (b actin) was taken as a loading control. **(d)** Real time PCRs for PDK1 and LDHA were performed. $**P < 0.01$, $***P < 0.001$ compared to si-Control (Hypoxia), $^{SS}P < 0.01$ compared to si-RASSF1+EV (Hypoxia) one-way ANOVA followed by SNK multiple comparison test. Data represent mean \pm s.e.m. $n = 3$ independent experiments from 3 biological replicates for human PSMCs.

To answer the other aspect of the question, regarding RASSF1C expression and its role under hypoxia, we carried out an array of further experiments. Unlike RASSF1A, PSMCs exposed to

24hr hypoxia, did not show any significant increase in RASSF1C mRNA expression (Supplementary Fig. 1a). This can be explained by distinctive promoter usage by RASSF1A and RASSF1C (PMID: 15867337), allowing for differential transcriptional regulation of both transcript variants.

Supplementary Fig. 1: (a) Human PSMCs were exposed to Normoxia (NOX) or Hypoxia (HOX) for 24 hr. Cell lysates were subjected to real time PCRs for RASSF1C. n = 3 independent experiments.

Further, RASSF1C overexpression did not have an effect on HIF-1 α protein expression and its transcriptional activity as observed by western blotting and HRE luciferase assay, respectively (Supplementary Fig. 6e, f).

Supplementary Fig. 6: (e) Human PSMCs were transfected with EV or RASSF1C-FLAG and 24 hr later, cells were exposed to hypoxia for another 24 hr. Cell lysates were subjected to western blotting for HIF1A, FLAG and ACTB. **(f)** A luciferase reporter under control of multiple HIF1 α binding sites (HRE) was transfected into cells with EV or RASSF1C-FLAG. 6 hr after transfection, cells were exposed to Hypoxia for 24 hr. Cells were lysed and luciferase activity was measured and normalized to co-transfected Renilla luciferase internal control. *** $P < 0.001$ compared to EV (Hypoxia), one-way ANOVA followed by SNK multiple comparison test. Data represent mean \pm s.e.m. n = 3 independent experiments.

Last but not least, no interaction was detected between HIF-1 α and RASSF1C under hypoxic conditions (Supplementary Fig. 8c) as seen in CoIP studies. Based on all these findings, we can conclude that the cross talk seen with HIF-1 α under hypoxia is specifically RASSF1A isoform driven (Fig. 1g-h, Fig. 4b-d, Fig. 5f-g of the revised manuscript).

c

Supplementary Fig. 8: (c) HEK 293 cells were transfected with RASSF1C-FLAG plasmid and exposed to hypoxia for 24 hr, followed by HIF1A IP and western blotting for HIF1A and FLAG. n = 2 independent experiments.

3. RASSF1A is a key regulator of hippo pathway signaling, yap/taz have also been implicated in HIF1 activation. This needs to be addressed to determine the full picture of the regulation.

R3: As rightly stated by the reviewer, RASSF1A is a well-documented regulator of hippo signaling. RASSF1A is shown to interact with MST1/2 kinases (upstream kinases of hippo signaling), displacing inhibitory RAF-1, resulting in dimerization and autophosphorylation of MST1/2. This leads to MST1/2 mediated phosphorylation of LATS kinases. LATS in turn phosphorylates transcriptional coactivator - YAP, leading to its cytoplasmic retention and proteasome mediated degradation (PMID: 25042563). In agreement with the literature, we also observed a decreased activity of YAP in presence of RASSF1A overexpression as shown by luciferase assay employing a TEAD luciferase reporter construct (Addgene ID: 34615). YAP and YAP mutant (S127A: constitutively active) overexpression was used as positive control (Supplementary Fig. 7a). In order to study the role of YAP in the RASSF1A mediated increase in HIF-1 α expression and activity, we overexpressed RASSF1A in presence of YAP wild type/mutant overexpression, followed by hypoxic exposure. Both YAP wild type and YAP mutant overexpression alone led to a further stabilization of HIF-1 α . However, it did not have any further effect on increased HIF-1 α stabilization observed under RASSF1A overexpression (Supplementary Fig. 7b). This finding was further substantiated by HRE luciferase assay where again YAP overexpression (wild type or mutant) did not show any additional effect on RASSF1A mediated HIF-1 α transcriptional activity (Supplementary Fig. 7c).

Supplementary Fig. 7: A luciferase reporter under control of (a) TEAD binding sites or (c) of multiple HIF1 α binding sites was transfected into HeLa cells with EV, YAP, YAP (S127A) and RASSF1A-FLAG. 6 hr after transfection, cells were exposed to Hypoxia for 24 hr. Cells were lysed and luciferase activity was measured and normalized to co-transfected Renilla luciferase internal control. *** $P < 0.001$ compared to EV (hypoxia), one-way ANOVA followed by SNK multiple comparison test. Data represent mean \pm s.e.m. (b) HEK 293 cells were transfected with plasmids indicated on top of lanes and exposed to hypoxia for 24 hr, followed by western blotting for HIF1A, YAP, RASSF1A and ACTB. $n = 3$ independent experiments.

Interestingly, several studies have reported hippo-signaling regulation under hypoxic exposure. Ma et al. found that hypoxia inactivated hippo signaling, which in turn enabled YAP-HIF-1 α interaction, further promoting hypoxia induced stabilization of HIF-1 α (PMID: 25438054). On similar lines, TAZ was reported as an interacting partner of HIF-1 α and positive regulator of its activity (PMID: 26059435). However, based on the above-mentioned results, we believe that the effect of RASSF1A on HIF-1 α expression and activity is independent of the hippo signaling pathway, and that the YAP/TAZ mediated regulation of HIF signaling as described in literature is a separate mechanism. The additional experiments referring to a putative contribution of RASSF1A-hippo pathway interaction to HIF1 activation are given in the revised manuscript.

4. Are RASSF1high associated with hypoxia signatures? Are these particularly aggressive or have a worse prognosis as would be expected of hypoxic tumors? From what I can determine it appears that the RASSF1 upregulation may be an important protective mechanism in coping with hypoxia, as HIF1 activation is not necessarily a dangerous event. However, is the absence of RASSF1A this may not function as well and lead to promotion?

R4. We thank the reviewer for the comment and based on it, we screened for mRNA expression of Lactate Dehydrogenase A (LDHA) and Carbonic anhydrase 9 (CA9) in the both, RASSF1^{high} and RASSF1^{low} tumors. LDHA and CA9 both are recognized biomarkers of cellular

hypoxia (PMID: 20461082; PMID: 281993910). Interestingly, fold increase in LDHA and CA9 expression compared to corresponding non-tumor parts was more pronounced in RASSF1^{high} tumors in comparison with RASSF1^{low} tumors (Fig. 8c of the revised manuscript). These results further support our hypothesis that RASSF1A potentiates hypoxia signaling.

Fig. 8: (c) RNA was isolated from tumor and matched non-tumor samples, followed by real time PCRs for indicated genes. The Ct values of tumor samples were divided by the Ct values of respective non-tumor samples to obtain the fold change of RASSF1A expression. * $P < 0.05$ compared to RASSF1A^{low}, unpaired Student's *t*-test. $n=9$ RASSF1A^{high} and $n=10$ RASSF1A^{low} tumors.

In order to establish a correlation if any between RASSF1A expression and clinical characteristics of the cancer patients, we increased the number of tumor samples with different pathological grades and screened them for RASSF1A protein expression. Similar to previous screening results, a percentage of tumors (22/56, 39%) displayed an increased RASSF1A expression compared to the matched non-tumor tissues (Supplementary Fig. 11a).

Supplementary Fig. 11: (a) Western blots for RASSF1A protein expression as analyzed in proteins isolated from tumor (T) and non-tumor (N) areas of human lung cancer. $n=57$ human lung tumor tissues with matched non-tumor tissues.

Interestingly, when comparing the lung cancer pathological stage with RASSF1A expression, we observed a significant increase in the expression of RASSF1A in stage III compared to stage I lung cancer patients (Fig. 8d). As it can be seen from the data, in nearly 50% of the stage III lung cancer patients a marked increase in RASSF1A expression was noted, which may suggest that RASSF1A^{high} lung cancers are particularly aggressive and/or have a worse prognosis via regulation of HIF-1 α and metabolic switch.

Fig. 8: (d) Fold change in RASSF1A expression in various lung tumor tissues plotted vs the pathological stages (I, II, III) of the respective tumors. * $P < 0.05$ compared to stage I, unpaired Student's t -test. $n=15$ stage I, $n=14$ stage II and $n=27$ stage III human lung tumor tissues.

5. In all, I really like the study and am supportive. I would encourage the authors to fully explain their findings in context (which do nicely fit) rather than being dismissive. Statements like 'Classified as a tumor suppressor, RASSF1A has predominantly been studied in cancer cell lines and transformed cells.' Implies everyone has been mistaken. There is far more clinical and in vivo data actually published than the work presented here. RASSF1A has been explored in HBECs and other normal cells, albeit not under hypoxia, but this displays a lack of informed opinion though not being aware of the literature.

The authors also write 'In contrast to several reports on decreased RASSF1A expression in a variety of tumors and tumor cell lines, some reports have indicated the lack of such decrease or even an increase in RASSF1A expression in subsets of tumors of various origin (Palakurthy et al., 2009; Pronina et al., 2012; Tezval et al., 2008).' Several reports? Try >1500 articles! This is not acceptable behaviour to increase the potential impact of your study. Especially we all you screened was '25 non-small cell lung tumor tissues', compared to the literal 10 of 1000s of samples. Again, this is unnecessary in an otherwise very nice study, and needs substantial amendment.

R5. We thank the reviewer for the support and critical suggestions to improve the manuscript. As mentioned, it was by no means our intention to dismiss the established role of RASSF1A as a tumor suppressor gene. Our intention is to report a hitherto unrecognized crucial role of RASSF1A in regulating HIF-1 α to promote hypoxia-driven gene regulation, metabolic switch and hyperproliferation in cells and tissues of pulmonary hypertension and in a subgroup of lung cancer patients. We believe that these are interesting findings and need to be explored further. As suggested by the reviewer, we corrected the statements that might otherwise

dismiss the tumor suppressive role of RASSF1A.

6. [Note from the Editor: reviewer #1 expressed in further correspondence that full western blots including molecular weight markers should be provided to confirm specific targeting of RASSF1A (point #2).]

R6. As requested by the reviewer, we have provided below the full western blot for the RASSF1A with samples treated with si-RASSF1 (Fig. X).

Fig. X: Western blot for RASSF1A protein. Human PSMCs were transfected with si-Control or si-RASSF1. 24hr later, cells were exposed to hypoxia for 24 hr. Cell lysates were subjected to western blotting for RASSF1A (ab23950: Abcam). RASSF1A band was observed at approximately 40kDa.

Reviewer #2 (Remarks to the Author):

In this manuscript, Dabral and colleagues report their observations on the role of RASSF1A-HIF1A in hypoxia-induced proliferation and glycolysis in cell cultures, in human tissues and a mouse model of hypoxic pulmonary hypertension (PH), and in human tissues and cell cultures of lung cancer specimens. The overall finding that RASSF1A strongly regulates HIF1A function and expression in hypoxia is a little outside the paradigm of direct hypoxic regulation of HIF1a, but the data are generally quite strong and this fits with an evolving literature of alternative regulators of HIF1a. Overall the manuscript is very well written and flows logically, with the figures clear.

Major

1. One of the key components in the work is in regards to how the phosphorylation of RASSF1A in the setting of hypoxia regulates RASSF1A stability and function, but this is presently not very well developed. What are the effects of RASSF1A phosphorylation on its stability over time (aside from direct expression levels as shown 2g), and the physical or functional interaction between it and HIF1a?

R1. We thank the reviewer for raising an important question on RASSF1A phosphorylation influence on it's own and on HIF-1 α stability. To study the effect of RASSF1A phosphorylation on its stability over time, we performed a cycloheximide (CHX, a protein synthesis inhibitor)

chase experiment. We treated HEK cells that overexpress RASSF1A wild type, S203A or S203D mutant with CHX (30µg/ml) for 1 hr and 3 hr, followed by cell lysis and screening for RASSF1A expression. We observed that the turnover of RASSF1A–S203A mutant was more rapid than that of RASSF1A wild type while the S203D mutant showed much slower turnover (Fig. 2g). Furthermore, to prove that the higher turnover of S203A mutant was dependent on proteasome-mediated degradation, the HEK cells overexpressing RASSF1A-S203A mutant were pretreated with MG132 (proteasome inhibitor), followed by CHX chase experiment. Treatment with MG132 successfully reversed the increased turn over of RASSF1A-S203A (Fig. 2h), leading to the conclusion that phosphorylation of RASSF1A at serine 203 residue increases its stability by protecting against proteasome mediated degradation.

Fig. 2: (g) HEK293 cells were transfected with plasmids indicated on top of lanes, treated with 30µg/ml cycloheximide (CHX), followed by hypoxia exposure for 1 hr and 3hr. Cell lysates were subjected to (g, upper) western blotting for RASSF1A and ACTB. (g, lower) Densitometrical quantified data of % of RASSF1A remaining. (h) HEK293 cells were transfected with RASSF1A-S203A mutant, pretreated with vehicle of MG132 (10µM) for 30min, followed by treatment with 30µg/ml CHX and hypoxia exposure for 1 hr and 3hr. Cell lysates were subjected to (h, upper) western blotting for RASSF1A and ACTB. (h, lower) Densitometrical quantified data of % of RASSF1A remaining. * $P < 0.05$, ** $P < 0.01$, *** $P < 0.001$ compared to RASSF1A-FLAG at the same time point or vehicle. $^{§§§}P < 0.001$ compared to RASSF1A-FLAG (S203A), Two-way ANOVA. Data represent mean \pm s.e.m. $n = 3$ independent experiments.

Further, to study the effect of RASSF1A phosphorylation on HIF-1α stability and function, we carried out a series of experiments. In PSMCs exposed to hypoxia, overexpression of RASSF1A-S203D mutant increased the expression of HIF-1α, similar to RASSF1A wild type while S203A mutant failed to show this effect (Fig. 4e). Consequently, S203A mutant did not lead to increase in HIF-1α transcriptional activity as measured by HRE luciferase assay (Fig. 4f).

Fig. 4: (e, left) Human PSMCs were transfected with EV, RASSF1A-FLAG, RASSF1A-FLAG (S203A) or RASSF1A-FLAG (S203D). 24 hr later, cells were exposed to Hypoxia or Normoxia for further 24 hr. Cell lysates were subjected to western blotting for HIF1A, RASSF1A and ACTB, followed by **(e, right)** densitometric analysis of relative HIF1A expression. **(f)** A luciferase reporter under control of multiple HIF1 α binding sites (HRE) was transfected into HeLa cells with EV, RASSF1A-FLAG, RASSF1A-FLAG (S203A) or RASSF1A-FLAG (S203D). 6 hr after transfection, cells were exposed to Hypoxia for 24 hr. Cells were lysed and luciferase activity was measured and normalized to co-transfected Renilla luciferase internal control. * $P < 0.05$, ** $P < 0.01$ compared to EV (Hypoxia), $^{\S}P < 0.05$, $^{\S\S}P < 0.01$ compared to RASSF1A-FLAG (S203A), one-way ANOVA followed by SNK multiple comparison test. Data represent mean \pm s.e.m. n = 3 independent experiments.

Lastly, there was a decreased interaction observed between HIF-1 α and RASSF1A-S203A mutant on pulling down HIF-1 α (Fig. 5e). Thus, RASSF1A phosphorylation at serine 203 residue regulates its functional and physical interaction with HIF-1 α .

Fig 5: (e) HEK293 cells were transfected with plasmids indicated on top of lanes and exposed to Hypoxia for 24 hr. HIF1A IP and RASSF1A co-IP were detected by western blotting. n = 2 independent experiments.

2. Can the authors determine the subcellular location of the expression (and ideally the site of its function) of RASSF1A in the cytoplasm versus nucleus? As noted, HIF1a stabilization primarily is a cytoplasmic event, with subsequent translocation to the nucleus and transcriptional function. One of the notable findings in the immunostaining (particularly the human tissue in figure 5b) is the diseased specimens have an increase in the relative expression of RASSF1 in the cytoplasm, but the nuclear expression of RASSF1 doesn't change much.

R2. To answer this question, we carried out sub-cellular fractionation of PSMCs exposed to different time points of hypoxia. RASSF1A was predominantly localized in the cytosolic fraction and increased on hypoxia exposure (Supplementary Fig. 8d).

Supplementary Fig 8: (d) Human PSMCs were exposed to Hypoxia or Normoxia for 12 hr and 24 hr, followed by subcellular fractionation. Cytoplasmic and nuclear lysates were subjected to western blotting for HIF1A, RASSF1A, LAMIN B1 and TUBA1A. Lamin B1 and alpha-tubulin (TUBA1A) were used as nuclear and cytoplasmic markers respectively. n = 2 independent experiments.

To further substantiate this observation, we performed a proximity ligation assay (PLA) using RASSF1A and HIF-1 α antibodies (Fig. 5g). Compared to Normoxic PSMCs, in PSMCs exposed to different time points of hypoxia, an interaction between RASSF1A and HIF1 α was observed. Interestingly, RASSF1A-HIF-1 α interaction was majorly confined to the cytosolic compartment (Fig. 5g), further proving that the HIF-1 α stabilization via RASSF1A interaction is majorly a cytoplasmic event.

Fig. 5: (g) Human PSMCs were exposed to hypoxia for 12 hr and 24 hr, followed by proximity ligation assay with HIF1A and RASSF1A antibodies. n = 3 independent experiments. Each red spot represents for a single interaction between HIF1A and RASSF1A and DNA was stained with DAPI (blue).

3. In the studies of the phenotype of the PSMCs from IPAH and donors (such as brdu uptake in figure 5f), it would ideal to perform dual knockdown of RASSF1A and HIF1a (as previously done in figure 2) to determine how much of the IPAH cell phenotype induced by RASSF1A is mediated by HIF1a.

R3: As suggested by the reviewer, we carried out dual knockdown of RASSF1 and HIF-1 α in IPAH PSMCs (similar to the experiment under hypoxia in donor PSMCs in Fig. 3c), followed

by BrdU incorporation assay to assess the effect on proliferation. The results are provided in Supplementary Fig. 9d of the revised manuscript. Similar to the results obtained in hypoxia-stimulated donor PSMCs (Fig. 3c), si-RASSF1 and si-HIF-1 α decreased proliferation of IPAH PSMCs equally, whereas a combination of both did not display any additional effect. Thus, the RASSF1A-HIF1a interplay appears to play a major role in this phenotype.

Supplementary Fig. 9: (d) IPAH PSMCs were transfected with RASSF1 siRNA (si-RASSF1), HIF1A siRNA (si-HIF1A) or in combination. 6 hr after transfection, cells were placed in medium with growth factors for 48 hr. Proliferation was measured by BrdU incorporation assay. $**P < 0.01$, $***P < 0.001$ compared to si-Control, one-way ANOVA followed by SNK multiple comparison test. Data represent mean \pm s.e.m. $n = 3$ independent experiments from 3 biological replicates for IPAH PSMCs.

4. Does knockdown of RASSF1 in the lung cancer cells affect the cell phenotype, such as rate of proliferation or BrdU incorporation?

R4. As requested by the reviewer, we carried out knockdown of RASSF1 in primary lung cells, followed by 48 hr hypoxia exposure, and proliferation was measure by BrdU incorporation. Interestingly, there was no change in proliferation observed in the tumor cells under hypoxia compared to normoxic control. Tumor cells inherently possess a strong proliferative capacity and therefore, hypoxia might not lead to a further increase in an otherwise high proliferation rate. However, knockdown of RASSF1 did result in a significant decrease in proliferation compared to si-Control. This results strongly suggest a role of RASSF1 in regulating the proliferative phenotype of these primary tumor cells. These data are provided in Supplementary Fig. 11d of the revised manuscript.

Supplementary Fig. 11: (d) Primary lung cancer cells were transfected with RASSF1 siRNA (si-RASSF1) or control siRNA (si-control). 24 hr after transfection, cells were placed in hypoxia for 48 hr. Proliferation was measured by BrdU incorporation assay. *** $P < 0.001$ compared to si-Control (hypoxia), one-way ANOVA followed by SNK multiple comparison test. Data represent mean \pm s.e.m. $n = 2$ independent experiments from 2 biological replicates, presented as separate.

Minor

5. It would be ideal, but is not critical, to performed immunostaining of the hypoxic PH mouse lung tissue for RASSF1, similar to the panel in figure 5b.

R5. As suggested by the reviewer, RASSF1A staining was carried out in lung sections from normoxic and 3 week hypoxia exposed mice. In corroboration with the *in vitro* findings, RASSF1 expression is increased in the remodeled pulmonary vasculature of 3 weeks hypoxic PH mouse lung tissue compared to normoxic mouse lung tissue (Supplementary Fig. 10).

Supplementary Fig. 10: RASSF1A is upregulated in hypoxic mouse lungs. Representative immunostaining microphotographs of mice lung sections from normoxic (n=3) and 3-week hypoxic mice (n=3), stained for RASSF1A (brown color). Scale bar: 20 μ m.

REVIEWERS' COMMENTS:

Reviewer #1 (Remarks to the Author):

I am satisfied with the additional experimental data and mostly OK with the rewarding, however, the discussion should include the authors opinions on how to explain the established prognostic value of RASSF1 null tumours in lung cancer - in light of this data. Please reference more upto date review on the clinical significance of RASSF1A - Grawenda et al. Brit J Cancer 2015.

In419. "Although majorly studied in the field of malignancies, studies on its potential role in primary cells under different physiological cues such as hypoxia are unexplored."

This is incorrect, Papaspyropoulos et al. Nat Comms 2018 described RASSF1 in ESC and early embryo, which importantly also included RNAseq data where HIF1 signalling was identified in shRNA of RASSF1. Please discuss this in context.

Reviewer #2 (Remarks to the Author):

I thank the authors for comprehensively addressing my requests and have no other concerns at this time.

REVIEWERS' COMMENTS:

Reviewer #1 (Remarks to the Author):

C1: I am satisfied with the additional experimental data and mostly OK with the rewarding, however, the discussion should include the authors opinions on how to explain the established prognostic value of RASSF1 null tumours in lung cancer - in light of this data. Please reference more upto date review on the clinical significance of RASSF1A - Grawenda et al. Brit J Cancer 2015.

R1. As suggested by the reviewer, the paper of Grawenda et al. Brit J Cancer 2015 has been cited. In addition, the opinion of authors on prognostic value of RASSF1 null tumours in lung cancer in light of our studies has been included in the revised version of the manuscript (page 12-13).

C2: In419. "Although majorly studied in the field of malignancies, studies on its potential role in primary cells under different physiological cues such as hypoxia are unexplored."

This is incorrect, Papaspyropoulos et al. Nat Comms 2018 described RASSF1 in ESC and early embryo, which importantly also included RNAseq data where HIF1 signalling was identified in shRNA of RASSF1. Please discuss this in context.

R2. This has been discussed in revised version of the manuscript (page 11).

Reviewer #2 (Remarks to the Author):

C1: I thank the authors for comprehensively addressing my requests and have no other concerns at this time.

R1: We would like to thank the reviewer for his valuable suggestions to improve the manuscript.